# Whole brain delivery of an instability-prone *Mecp2* transgene improves behavioral and molecular pathological defects in mouse models of Rett syndrome

Mirko Luoni[1,2†], Serena Giannelli[1†], Marzia Tina Indrigo[1], Antonio Niro[1,2], Luca Massimino[1], Angelo Iannielli[1,2], Laura Passeri[3], Fabio Russo[3], Giuseppe Morabito[1], Piera Calamita[4], Silvia Gregori[3], Benjamin Deverman[5], Vania Broccoli[1,2]*

[1]Stem Cell and Neurogenesis Unit, Division of Neuroscience, San Raffaele Scientific Institute, Milan, Italy; [2]CNR Institute of Neuroscience, Milan, Italy; [3]San Raffaele Telethon Institute for Gene Therapy (SR-Tiget), San Raffaele Scientific Institute IRCCS, Via Olgettina, Milan, Italy; [4]National Institute of Molecular Genetics (INGM), Milan, Italy; [5]Stanley Center for Psychiatric Research at Broad Institute, Cambridge, United States

**\*For correspondence:**
broccoli.vania@hsr.it

†These authors contributed equally to this work

**Competing interests:** The authors declare that no competing interests exist.

**Abstract** Rett syndrome is an incurable neurodevelopmental disorder caused by mutations in the gene encoding for methyl-CpG binding-protein 2 (MeCP2). Gene therapy for this disease presents inherent hurdles since *MECP2* is expressed throughout the brain and its duplication leads to severe neurological conditions as well. Herein, we use the AAV-PHP.eB to deliver an instability-prone *Mecp2* (i*Mecp2*) transgene cassette which, increasing RNA destabilization and inefficient protein translation of the viral *Mecp2* transgene, limits supraphysiological Mecp2 protein levels. Intravenous injections of the PHP.eB-iMecp2 virus in symptomatic *Mecp2* mutant mice significantly improved locomotor activity, lifespan and gene expression normalization. Remarkably, PHP.eB-iMecp2 administration was well tolerated in female *Mecp2* mutant or in wild-type animals. In contrast, we observed a strong immune response to the transgene in treated male *Mecp2* mutant mice that was overcome by immunosuppression. Overall, PHP.eB-mediated delivery of i*Mecp2* provided widespread and efficient gene transfer maintaining physiological Mecp2 protein levels in the brain.

## Introduction

Rett syndrome (RTT) is a severe neurological disorder and second cause of intellectual disabilities in girls. RTT is distinguished by a period of 6–12 month of overtly normal development followed, then, by a rapid regression with the loss of the purposeful motor skills and the onset of repetitive and autistic behaviors (*Lombardi et al., 2015*; *Katz et al., 2016*). In the vast majority of cases RTT is caused by loss-of-function mutations in the *MECP2* gene, which encodes for the methyl-CpG binding protein 2 (MeCP2), a global chromatin regulator highly expressed in neurons (*Bienvenu and Chelly, 2006*). Neuropathological studies have shown that RTT brains exhibit abnormal neuronal morphology, but no sign of neuronal death. *Mecp2*-deficient mice recapitulate key neurological deficits observed in RTT patients offering an invaluable model where to investigate pathological mechanisms as well as test innovative therapies (*Guy et al., 2001*). Similar to human patients, RTT mice show

microcephaly without gross neuropathological changes or neurodegeneration. In contrast, recent studies have convincingly revealed that the *Mecp2* loss significantly alters neuronal activity leading to a progressive imbalance of the excitatory-inhibitory synaptic activity across the brain with divergent modalities occurring between different circuits and regions of the brain (*Durand et al., 2012*; *Banerjee et al., 2016*). In 2007, a seminal study by the Adrian Bird's group demonstrated that the RTT pathological phenotype can be significantly reversed in mice by re-activating the *Mecp2* even at advanced disease stages (*Guy et al., 2007*). In fact, genetic reactivation of *Mecp2* in more than 70% of the neurons in adult mice normalized brain morphology and significantly improved several sensory-motor dysfunctions (*Robinson et al., 2012*). These findings provide a strong evidence that Mecp2 is a key factor in maintaining full neurological function during adulthood. Consistently, multiple pathological manifestations exhibited by adult mutant RTT mice can be fully recapitulated by deleting *Mecp2* exclusively in adulthood (*Cheval et al., 2012*; *McGraw et al., 2011*; *Nguyen et al., 2012*).

Despite the proven genetic reversibility of the RTT disease phenotype in mice, translational treatments aiming at curing the disease or some of its neurological symptoms have not been successful yet. In fact, MeCP2 is a global determinant of the neural chromatin structure and is a pervasive regulator of gene expression in brain cells and, thus, it remains challenging to target a single MeCP2 downstream pathway to obtain a substantial therapeutic benefit (*Lombardi et al., 2015*; *Katz et al., 2016*). The inherent monogenic nature of RTT makes gene therapy a strong translational option for this disease. However, *MECP2* gene duplication in humans is responsible for a serious and clinically distinguished neurodevelopmental disorder. Affected males present with early hypotonia, limb spasticity and severe intellectual disability (*Miguet et al., 2018*; *Van Esch et al., 2005*). Thus, a successful gene therapy for RTT has to deliver the correct MeCP2 dosage in a tight range that overlaps with endogenous levels. Different studies have recently provided encouraging results showing that intravenous administration of an Adeno-associated virus serotype 9 (AAV9) expressing the wild-type (WT) *MECP2* attenuated neurological dysfunctions and extended lifespan in RTT mice (*Gadalla et al., 2013*; *Gadalla et al., 2017*; *Garg et al., 2013*; *Matagne et al., 2017*). However, the AAV9 does not cross efficiently the blood-brain barrier in adult mice while having a preferential tropism for peripheral organs. In fact, intravenous delivery of high doses of AAV9 can lead to severe liver toxicity and sudden death due to ectopic MeCP2 expression (*Gadalla et al., 2017*). Moreover, the limited brain transduction obtained in these studies was not sufficient to determine a correlation between the viral dose, transduction efficiency and therapeutic benefits. Finally, it remains to be determined whether a gene therapy approach is capable to rescue molecular dysfunctions and transcriptional alterations caused by loss of *Mecp2* in the adult brain.

Recently, novel AAV9 synthetic variants have been generated through selected mutagenesis of the capsid proteins to enhance gene transfer in the brain after viral systemic delivery. In particular, using a cell type-specific capsid selection screening, the novel engineered capsid AAV-PHP.B has been developed through a 7-amino-acid insertion in the AAV9 VP1 capsid protein (*Deverman et al., 2016*). The AAV-PHP.B viral particles are able to permeate the blood-brain barrier of adult mice and diffuse throughout the neural parenchyma transducing with high efficiency both neurons and glia (*Deverman et al., 2016*; *Morabito et al., 2017*). A subsequent enhanced version has been isolated, named AAV-PHP.eB (PHP.eB in short), with increasing efficiency in brain targeting, thus, requiring lower doses of virus for high neuronal transduction (*Chan et al., 2017*). We sought to use the PHP.eB as viral platform to sustain a global and efficient *Mecp2* brain transduction in adult mutant and WT mice. This system provided us with a flexible and potent tool to determine the efficacy and safety of diffuse *Mecp2* gene transfer in the adult mouse brain. Herein, using PHP.eB transductions in neuronal cultures, we characterized the viral gene expression cassette in all its elements to define its overall transcripts instability and limited translational efficiency. For this reason, the use of a constitutive strong promoter resulted more suitable to restore Mecp2 protein levels in a physiological range and limit viral overload, without compromising its disease rescue efficacy. Indeed, we obtained a robust phenotypic amelioration both in male and female *Mecp2* mutant mice, without severe toxicity effects in wild-type animals. In addition, we identified and characterized a strong immune response to the exogenous Mecp2 in male mutant mice, that severely affected the life-span of treated animals. This limitation was overcome by chronic immunosuppression that granted a life-span extension of up to nine months.

## Results

### Designing an instable *Mecp2* transgene cassette with reduced translation efficiency

Given that the therapeutic range of *Mecp2* gene expression is very limited, earlier studies employed a viral cassette with a minimal *Mecp2* promoter with the attempt to reconstitute endogenous gene expression levels. However, the regulatory regions of *Mecp2* remain poorly characterized and the use of self-complementary AAV9 with a limited cargo capacity has obliged to include only very short fragments of the proximal promoter (*Gadalla et al., 2017*; *Garg et al., 2013*; *Matagne et al., 2017*; *Sinnett et al., 2017*). With these conditions, it remains very unlikely to recapitulate the endogenous *Mecp2* expression pattern, whereas the risk of insufficient expression becomes higher. More generally, the lack of a system to compare exogenous with endogenous levels of MeCP2 is preventing the optimization of the viral transgene cassette. Herein, we overcame these limitations by setting a novel validation assay to compare the transcriptional efficiency of different viral *Mecp2* transgene cassettes directly in neuronal cultures. We initially confirmed that more than 90% of primary mouse neurons can be transduced with the PHP.eB virus. Then, we generated a transgene cassette with the *Mecp2_e1* isoform including the coding sequence and a short proximal 3'UTR (3'pUTR,~200 bp). This *Mecp2* transcript occurs naturally in embryonic stem cells and various tissues, but during development of the neural system this form is progressively replaced by transcripts with longer 3'-UTR (8,6 Kb) (*Rodrigues et al., 2016*). *Mecp2* transgene sequence carrying an N-terminal V5 tag was driven by two types of promoter (*Figure 1a*). We opted to use the chicken-β-actin promoter, which has been extensively used in gene therapy for driving high levels of expression in both neurons and glial cells (M2a, *Figure 1a*). Alternatively, we cloned a 1.4 kb fragment of the *Mecp2* promoter (that following previous studies includes the vast majority of regulatory regions identified so far) with the intent to recapitulate the endogenous *Mecp2* expression pattern (M2b, *Figure 1a*). *Mecp2*$^{-/y}$ primary neurons were infected with either viruses. As control we used wild-type (WT) neurons infected with a GFP vector (GFP, *Figure 1a*), and two weeks later were lysed for immunoblotting and genome copy quantification. Surprisingly, using this *Mecp2* promoter fragment, the total viral Mecp2 protein amount remained significantly lower respect to endogenous levels in transduced neurons (*Figure 1b*). Conversely, mutant neurons infected with the cassette containing the CBA promoter exhibited a physiological range of total Mecp2 protein. Moreover, similar results were obtained using Mecp2 immunofluorescence staining on infected neuronal cultures (*Figure 1c*). This difference was not depended on the relative infection load, since viral copy numbers were equivalent in neurons transduced with either of the two viruses (*Figure 1d*). Thus, only the use of the CBA strong promoter was capable of re-establishing Mecp2 protein levels in a physiological range without exceeding viral doses. We, then, asked why and through which mechanism the CBA-*Mecp2* cassette (M2a) could not exceed the endogenous Mecp2 protein range despite the use of a strong promoter and the presence of multiple copies of the virus in the neurons (*Figure 1d*). We speculated that the lack of the 5'- and 3'-UTR complete sequences from the *Mecp2* transgene might intrinsically impair protein production. During neuronal development, *Mecp2* transcripts with a long 3'-UTR are highly stabilized leading to progressive Mecp2 protein accumulation (*Rodrigues et al., 2016*). In contrast, alternative *Mecp2* isoforms with shorter 3'-UTRs are less stable and poorly regulated during development. Thus, we sought to compare the relative stability of the viral *Mecp2* transcript respect to the total endogenous *Mecp2* mRNA by measuring its half-life after gene transcription arrest with Actinomycin D (ActD) (*Figure 1e*). Using the same qRT-PCR primers and reaction, viral *Mecp2* transcripts were selectively amplified from PHP.eB-*Mecp2* transduced neuronal cultures isolated from *Mecp2* knock-out (KO) embryos, while total endogenous *Mecp2* mRNAs were obtained from WT neuronal cultures (*Figure 1f*). Remarkably, viral *Mecp2* transcripts showed significant lower RNA levels respect to the total endogenous *Mecp2* transcripts after 300 min of ActD treatment (51 ± 4%) (*Figure 1f*). Thus, the lack of a long 3'-UTR generates an instability-prone *Mecp2* (i*Mecp2*) isoform which is significantly destabilized in neuronal cultures. To determine if this reduction in RNA half-life was only determined by the pUTR, we generated an additional cassette with a synthetic assembled 3'-UTR (aUTR) which merged most of the known regulatory elements scattered in the 8,6 Kb long 3'-UTR as previously reported (*Matagne et al., 2017*) (M2c, *Figure 1a*). In addition, we generated two more vectors, one including the physiological 5'-UTR sequence in combination with the pUTR sequence (M2d, *Figure 1a*) and a second lacking the 3'-UTR thus carrying only the polyA (pA)

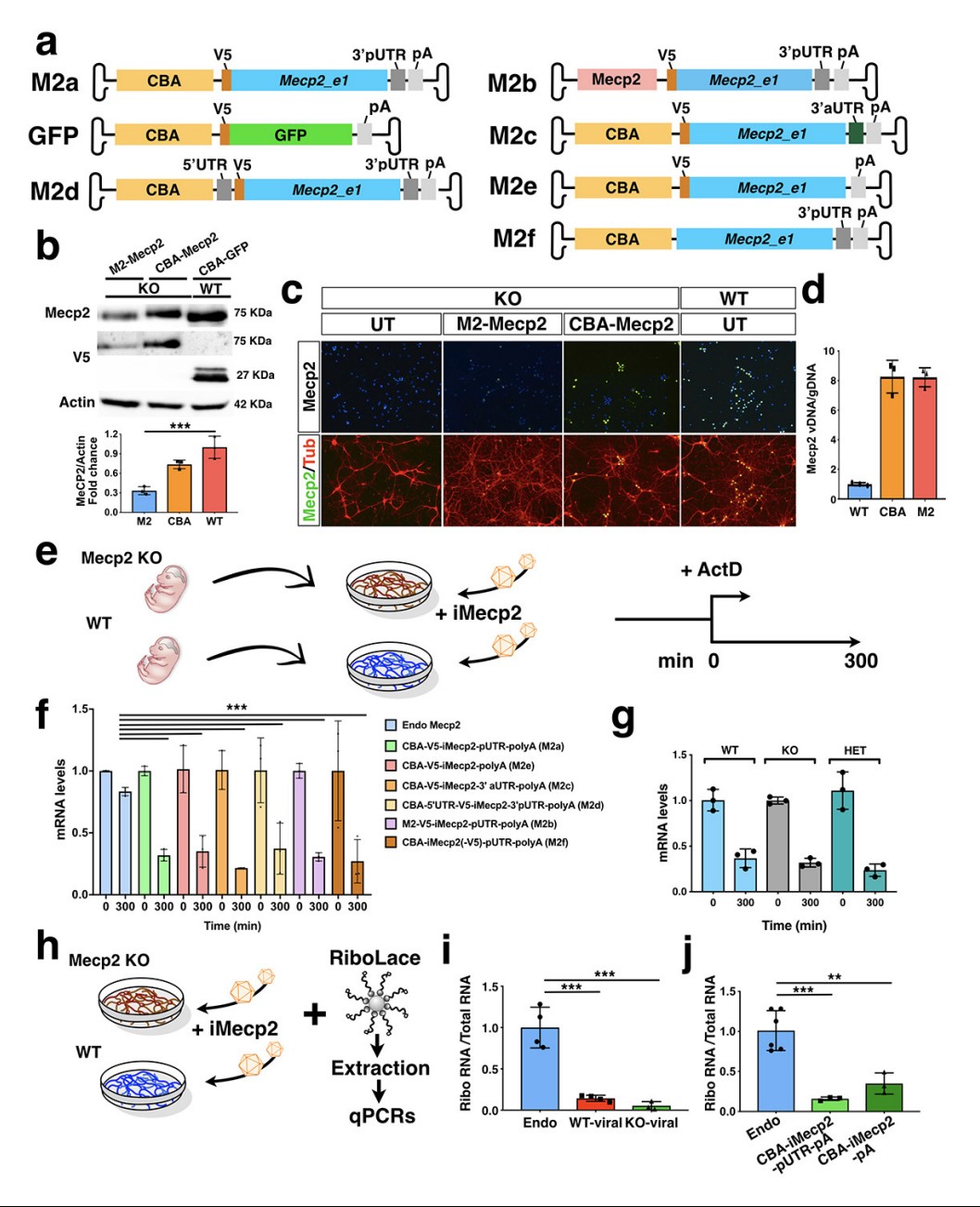

**Figure 1.** RNA stability and translational efficiency of the viral *Mecp2* transgene. (a) Illustration of the AAV vectors expressing V5-tagged *Mecp2_e1* or GFP under the control of Chicken β-Actin (CBA) promoter or *Mecp2* core promoter. Vectors included murine *Mecp2* coding sequence with either its own proximal 3'UTR (3'pUTR, M2a), or a synthetic 3'UTR sequence (3'aUTR, M2c), or both 3'pUTR and 5'UTR (M2d), or no UTR (M2e), or without the V5 tag (M2f). (b) Western blot analysis for V5, Mecp2 and Actin protein levels in GFP infected (control, CBA-GFP) WT neurons and *Mecp2⁻/ʸ* (KO) neurons infected with CBA-*Mecp2* and M2-*Mecp2*. Quantification was performed using densitometric analysis of Mecp2 relative to Actin signal and expressed in arbitrary units (n = 3) (c) Immunostaining of KO (control untreated and infected with M2-*Mecp2* or CBA-*Mecp2*) and wild-type neurons for Mecp2 and TUBB3 (Tub). (d) qRT-PCR quantification of viral *Mecp2* DNA copies in Mecp2 KO neurons relative to genomic DNA (n = 3). (e) Illustration of the experimental work-flow to study the RNA stability in KO and WT neurons (f) RNA stability of endogenous (in WT neurons) and viral (from the 5 different vectors described above in KO neurons) *Mecp2* transcript determined by qRT-PCRs (n = 3, t = 300 min were normalized over t = 0 values and compared among different treatments). (g) RNA stability of viral (CBA-iMecp2-pUTR-pA) *Mecp2* transcript in neurons derived from different genotypes: WT, KO and *Mecp2⁺/⁻* (Het) determined by qRT-PCRs (n = 3, t = 300

*Figure 1 continued on next page*

*Figure 1 continued*

min were normalized over t = 0 values and compared among different treatments) (**h**) Illustration of the experimental work-flow to study translational efficiency. (**i**) qRT-PCR of viral and endogenous *Mecp2* RNA in the ribosomal fraction normalized on the total RNA in WT and KO neurons (n = 4 endogenous *Mecp2*, n = 4 exogenous *Mecp2* in KO neurons, n = 3 exogenous *Mecp2* in WT neurons). (**j**) qRT-PCR of viral and endogenous *Mecp2* RNA in the ribosomal fraction normalized on the total RNA in WT (n = 6) and KO neurons infected with 2 different viral Mecp2 construct with (CBA-iMecp2-pUTR-pA, n = 3) and without (CBA-iMecp2-pA, n = 3) the 3'-pUTR. Error bars, Standard Deviation (SD). **p<0.01, ***p<0.001, compared groups are indicated by black lines. ANOVA-one way, (**b, f, g, i, j**) and Tukey's post hoc test. Scale bar: 100 μm (**c**).

sequence (M2e, *Figure 1a*). *Mecp2* RNA levels in neurons transduced with these viral vectors remained very unstable over time compared to endogenous mRNA levels (M2e: 48 ± 13%; M2c: 62% ± 0,2%; M2d 46 ± 20%; M2b 53 ± 3%). Finally, we generated and tested a cassette lacking the V5 tag, to exclude a possible interference on the transgene mRNA stability (M2f, *Figure 1a*), without observing significant differences in comparison with previous configurations (73 ± 18%, *Figure 1f*). Thus, our results confirmed that Mecp2 gene expression is regulated by complex mechanisms and multiple factors, especially at RNA level. This complexity can hardly be restricted in the limited capacity of an AAV vector but is essential for the design of future *Mecp2* gene replacement strategies. *Mecp2* mRNA instability was independent by the genotype of the transduced neurons since similar mRNA loss was detected in WT, *Mecp2* mutant and heterozygote neurons (*Figure 1g*).

Next, we sought to determine the translational efficiency of the *Mecp2* viral variant. For this aim, we employed the RiboLace, a novel methodology based on a puromycin-analog which enables the isolation of the ribosomal fraction in active translation with their associated RNAs (*Clamer et al., 2018*). Thus, the translational active ribosomal fraction was captured by RiboLace-mediated pull-down from lysates of uninfected WT or PHP.eB-iMecp2 transduced *Mecp2*-KO neuronal cultures (*Figure 1h*). Subsequently, mRNAs were extracted from both the isolated ribosomal fractions and the total lysates and used for RT-qPCR analysis with the same set of *Mecp2* primers. Remarkably, the normalized fraction of viral *Mecp2* transcripts associated with translating ribosomes was reduced by 83 ± 5% compared with that of ribosome-bound endogenous *Mecp2* mRNA (*Figure 1i*). A similar reduction was calculated by using a viral *Mecp2* with or without the pUTR suggesting that this impairment is independent by the associated non-coding elements (*Figure 1j*). Next, the relative stability of the Mecp2 proteins produced by either the endogenous or the viral gene were assessed in neurons after treatment with cycloheximide (CHX). No difference in Mecp2 protein levels were found up to 8 hr after CHX administration (*Figure 2a*). Finally, we asked whether *MECP2* RNA instability can represent a hurdle also in designing viral cassettes with the human *MECP2* gene. Thus, we employed a pair of male isogenic iPSC lines either control or with a CRISPR/Cas9 induced *MECP2* loss-of-function mutation (*Figure 2b,c*). Both iPSC lines were differentiated in vitro into cortical neuronal cultures and, then, control and *MECP2* mutant lines were transduced with AAV expressing either GFP or the human version of i*MECP2*, respectively (*Figure 2d,e*). Noteworthy, viral *MECP2* transcript stability was severely affected as compared to endogenous RNA levels to an extent comparable to what observed with murine mRNAs (*Figure 2f*).

In summary, these data revealed the crucial role of the post-transcriptional processes in determining the final output of the viral Mecp2 protein levels. Given this limited efficiency in transcript stability and translation efficacy only the use of the CBA strong promoter was effective to sustain Mecp2 protein levels comparable to those found in WT neurons. Since all the viral cassettes caused a similar instability of *Mecp2* transcripts, among them we chose CBA-i*Mecp2*-pUTR-pA (M2a) construct for further in vivo studies, and referred to it as i*Mecp2*, for instability-prone *Mecp2*.

## Efficient *iMecp2* gene transfer throughout the brain of adult male *Mecp2* mutant mice

To test the efficacy of gene transfer with the i*Mecp2* transgene cassette we designed a dose escalation approach to administer 10-fold increasing doses of PHP.eB-iMecp2 ($1 \times 10^9$ vg, $1 \times 10^{10}$ vg, $1 \times 10^{11}$ vg and $1 \times 10^{12}$ vg/mouse) of virus through intravenous delivery in 4 weeks old $Mecp2^{-/y}$ mice and untreated $Mecp2^{-/y}$ animals were utilized as controls (*Figure 3—figure supplement 1a*). To determine the exact brain penetration efficiency and neural tissue transduction of the PHP.eB-

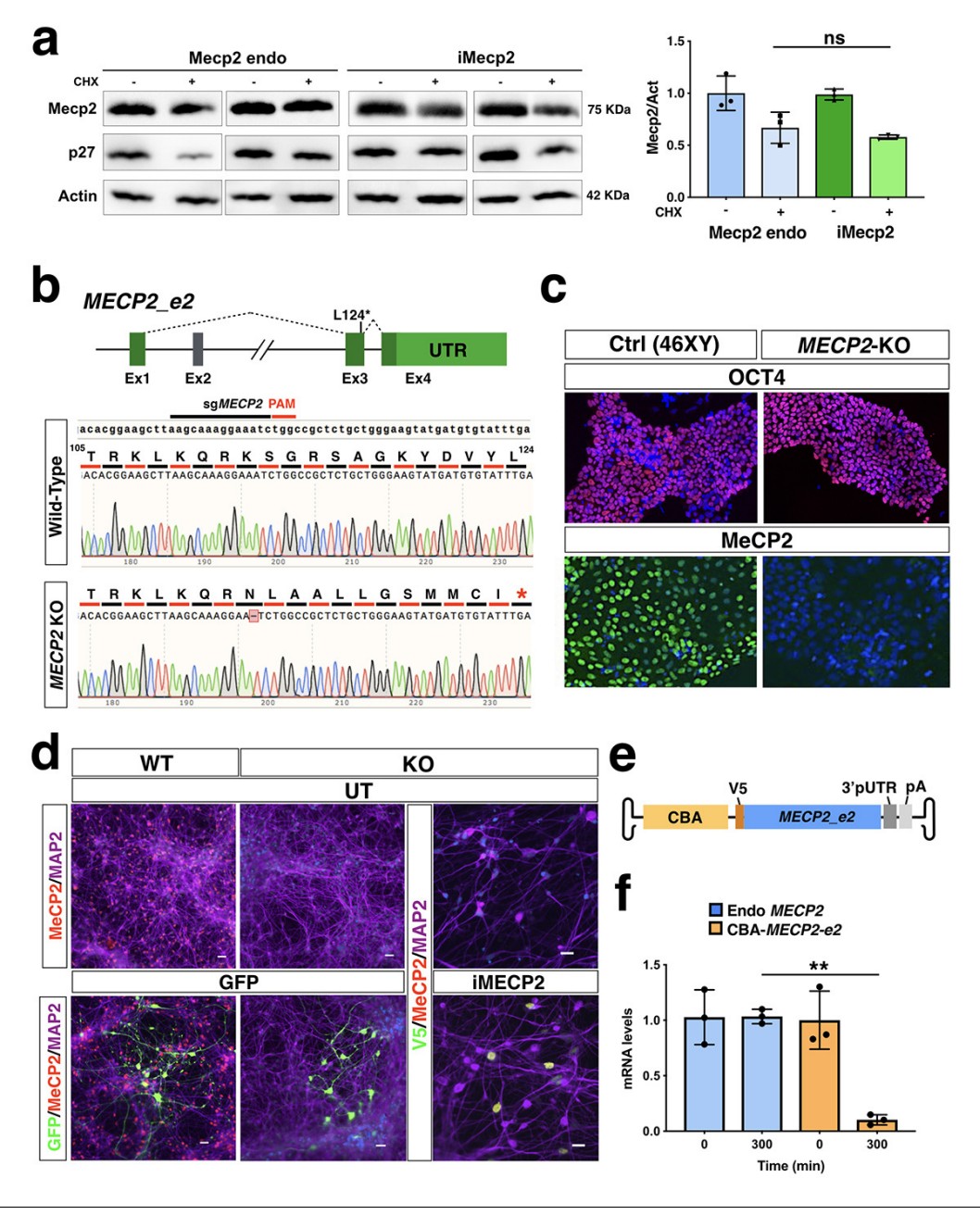

**Figure 2.** Mouse protein stability and stability of the viral human i*MECP2* construct in iPSC-derived cortical neurons. (**a**) Protein stability was assessed in murine neurons using western blot analysis to quantify Mecp2 and p27 protein reduction after Cycloheximide treatment (50 uM, CHX, 8 hr) over Actin protein and corresponding densitometric quantification expressed in arbitrary units (right panel) (n = 3). t = 8 hr were normalized over t = 0 values and compared among different treatments (ns = p > 0.05, ANOVA-one way, Tukey's post hoc test). (**b**) Illustration of the CRISPR-Cas9 based strategy for genetic editing of male control human iPSCs to obtain *MECP2* KO cells. The sgRNA selected for this approach annealed on exon 3 of the *MECP2* gene and generated a single nucleotide deletion. That resulted in a frameshift of its coding sequence and a premature STOP codon 12 residues downstream. (**c**) Human iPSCs carrying this mutation (*MECP2* KO) maintained their pluripotency marker Oct4 but did not presented detectable MeCP2 protein as tested by immunofluorescence when compared to control cells (Ctrl). (**d**) WT and *MECP2*[-/y] iPSCs were successfully differentiated into neurons (MAP2 staining) and transduced using PHP.eB vectors carrying either GFP or i*MECP2*. Immunofluorescence respectively for GFP and MeCP2 attested neuronal transduction. (**e**) Schematics of the AAV vector used for RNA stability experiment and expressing V5-tagged *MECP2_e2* under the control of Chicken β-Actin (CBA) promoter. (**f**) *MECP2* RNA stability

*Figure 2 continued on next page*

*Figure 2 continued*

was determined by qRT-PCRs in WT neurons, to test the endogenous transcript (blue bars, n = 3), and in KO neurons infected with the aforementioned vector, to test the viral transcript (yellow bars n = 3). t = 300 min were normalized over t = 0 values and compared among different treatments. UT: untreated.

iMecp2, mice for each viral dose were euthanized and brains separated in two halves for immunohistochemistry and immunoblot analysis two weeks after administration. Sections of the infected *Mecp2* mutant brains were stained for Mecp2 or V5 to visualize the viral Mecp2 transduction pattern. The distribution of PHP.eB-iMecp2 was spread throughout the brain with increasing intensity using higher viral doses (*Figure 3b*), as also measured with fluorescence intensity (*Figure 3—figure supplement 1b*). Moreover, sub-cellular analysis of viral Mecp2 protein distribution confirmed a strong enrichment in nuclear heterochromatic foci, mirroring the genome-wide distribution of endogenous Mecp2 (*Figure 3—figure supplement 1c*). Quantification of Mecp2$^+$ cells respect to DAPI$^+$ nuclei in the somatosensory cortex and striatum showed a proportional increase in transduction efficiency from the lowest ($1 \times 10^9$ vg; 15 ± 3% in cortex; 18 ± 3% in striatum) to the highest dose ($1 \times 10^{12}$ vg; 78 ± 3% in cortex; 80 ± 4% in striatum). Brain tissues transduced with $10^{11}$ and $10^{12}$ vg of PHP.eB-iMecp2 and stained for neural cell type specific markers (NeuN and Sox9, that stain neurons and astrocytes, respectively) showed that viral transduction was efficient (above 50%) in infecting both neuronal and glial cells, respectively (*Figure 3—figure supplements 2* and *3*). Moreover, cortical GABAergic interneurons, whose dysfunction is a crucial determinant of the RTT phenotype, were effectively transduced (*Figure 3—figure supplement 2c*; *Chao et al., 2010*). Whereas, at lower dosages $10^9$ and $10^{10}$ vg, neurons were far less transduced in comparison with astrocytes, whose transduction efficiency was similar among the different groups, indicating a higher tropism for these cells (*Figure 3—figure supplement 3* and *Supplementary file 1*).

Subsequent western blot analysis revealed increasing total levels of Mecp2 protein in cortical and striatal tissues upon transduction with higher doses of PHP.eB-iMecp2 (*Figure 3d*). In particular, administration of $10^{11}$ vg of PHP.eB-iMecp2 resulted in Mecp2 protein levels comparable to those detectable in control brains (*Figure 3d*). Conversely, $10^{12}$ vg of PHP.eB-iMecp2 triggered a 3-fold increase in *Mecp2* expression respect to endogenous levels (*Figure 3d*).

To further assess the efficiency of viral transduction, we measured the number of viral i*Mecp2* copies present in the brain and liver of the treated mice. PHP.eB-iMecp2 treated animals showed a higher number of viral copies in the brain respect to the liver (brain: 15 ± 5, $10^{11}$ vg; 65 ± 15, $10^{12}$ vg. liver: 7 ± 2, $10^{11}$ vg; 55 ± 15, $10^{12}$ vg) (*Figure 3—figure supplement 4a*) confirming that the PHP.eB capsid has higher propensity to transduce the neural tissue respect to peripheral organs (*Chan et al., 2017*). In contrast, transgene RNA levels were proportionally less abundant compared to the relative viral genome copy in brain respect to the liver (*Figure 3—figure supplement 4a*). In addition, despite the significant increase in i*Mecp2* genomic copies and total mRNA, protein levels were only marginally augmented in brain and partially in liver (*Figure 3d* and *Figure 3—figure supplement 4a*). This effect was particularly evident in cortical brain samples with the $10^{12}$ vg dose that compared to wild-type triggered an increase of 80-fold in mRNA, but only of 3-fold in protein. These observations confirmed that i*Mecp2* mRNA is poorly transcribed and translated in brain tissue as previously shown in neuronal cell cultures. Altogether, these data clearly highlight the robust efficiency of the PHP.eB capsid to cross the blood-brain barrier in adult mouse brains and to spread throughout the neural tissue transducing large number of cells. Importantly, the four doses of PHP.eB-iMecp2 which differed by a 10-fold higher titer showed a proportional increase in transduction efficiency in the brain. Hence, this escalation in viral transduction offered a great opportunity to test the extent of phenotypic rescue in *Mecp2* mutant mice depending by the viral gene transfer efficiency and the relative number of cells with restored *Mecp2* expression.

## Severe immune response to exogenous i*Mecp2* and its suppression by cyclosporine in transduced *Mecp2*$^{-/y}$ mice

Next, we treated mice with increasing doses of PHP.eB-iMecp2 ($1 \times 10^9$ vg, $1 \times 10^{10}$ vg, $1 \times 10^{11}$ vg and $1 \times 10^{12}$ vg) and their control littermates (WT and GFP treated *Mecp2*$^{-/y}$ mice, $1 \times 10^{11}$ vg). Four weeks old *Mecp2*$^{-/y}$ mice were intravenously injected and examined over time to monitor the progression of behavioral deficits and the relative efficacy of the treatments. As previously reported,

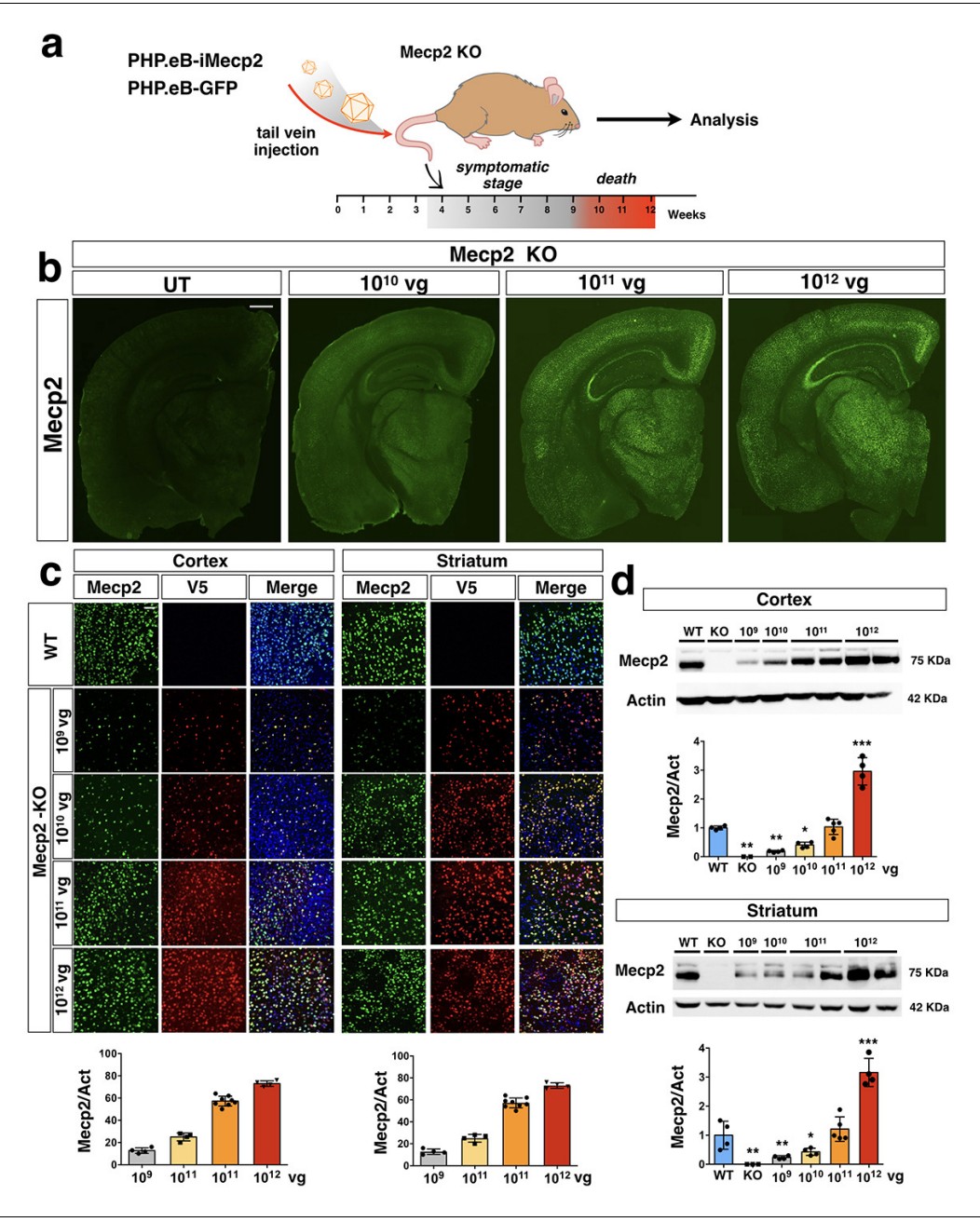

**Figure 3.** PHP.eB-mediated i*Mecp2* gene transfer in symptomatic *Mecp2⁻/y* mouse brains. (a) Illustration of the experimental setting to restore the expression of *Mecp2* in symptomatic mutant animals by AAV-PHP.eB systemic transduction. (b) Low magnification of Mecp2 immunostaining in brains of KO control untreated (UT) and treated animals ($1×10^{10}$, $1×10^{11}$, $1×10^{12}$ vg/mouse). (c) High magnification immunostaining for Mecp2 and V5 in cortex and striatum derived from wild-type (WT) and Mecp2 treated KO ($1×10^{9}$, $1×10^{10}$, $1×10^{11}$, $1×10^{12}$ vg/mouse) animals (*Mecp2*-KO). Nuclei were stained with DAPI (merge panels). Bottom panel: bar graphs showing the fraction of Mecp2 positive on the total DAPI positive (n = 4 for $1×10^{9}$–$1×10^{10}$–$1×10^{12}$; n = 8 $1×10^{11}$ vg/mouse). (d) Western blot analysis to quantify Mecp2 over Actin protein levels in cortex (upper panel) and striatum (lower panel) derived from WT, untreated KO and i*Mecp2* treated K*O* ($1×10^{9}$, $1×10^{10}$, $1×10^{11}$, $1×10^{12}$ vg/mouse) animals and corresponding densitometric quantification expressed in arbitrary units (n = 4 for $1×10^{9}$–$1×10^{10}$–$1×10^{12}$; n = 5 $1×10^{11}$ vg/mouse). Error bars, SD. *$p<0.05$, **$p<0.01$ and ***$p<0.001$ as compared to WT mice (ANOVA-one way with Tukey's post hoc test). Scale bars: 500 μm (b), 20 μm (c).

The online version of this article includes the following figure supplement(s) for figure 3:

**Figure supplement 1.** Distribution and quantification of i*Mecp2* gene transfer in *Mecp2⁻/y* brains.

*Figure 3 continued on next page*

*Figure 3 continued*
**Figure supplement 2.** Analysis of transduced neurons after i*Mecp2* gene transfer in *Mecp2*$^{-/y}$ brains.
**Figure supplement 3.** Analysis of transduced astrocytes after i*Mecp2* gene transfer in *Mecp2*$^{-/y}$ brains.
**Figure supplement 4.** Characterization of i*Mecp2* transgene expression in brain and liver of *Mecp2*$^{-/y}$ animals.

at the age of injection *Mecp2*$^{-/y}$ mice in a C57Bl/6 background had already reduced body size and were lighter compared to their WT littermates and between 9–11 weeks of age they experience a sudden weight loss which anticipated the worsening of RTT symptoms and their following decease (*Guy et al., 2001*; *Figure 4a*). Animals were euthanized upon a body weight loss of 20% for ethical reasons. In absence of treatment *Mecp2*$^{-/y}$ presented with these features already at 5–6 weeks therefore treatment was never delivered in mice older than this age. A similar lifespan length was observed in mice administrated with two different doses of control PHP.eB-GFP virus ($10^{11}$ vg and $10^{12}$ vg). Conversely, the *Mecp2*$^{-/y}$ mice treated with $10^{10}$ vg and $10^{11}$ vg of PHP.eB-iMecp2 showed a significant increase in lifespan reaching a median survival of 59d and 68d, respectively (*Figure 4a*). These beneficial effects were not observed in *Mecp2*$^{-/y}$ mice injected with the lowest dose of PHP.eB-iMecp2 ($10^{9}$ vg) whose lifespan remained comparable to that of control treated animals (*Figure 4a*). Unexpectedly, all the animals (n = 6) exposed to the $10^{12}$ vg dose of PHP.eB-iMecp2 died within two weeks from the viral administration and, then, excluded from the following functional tests (*Figure 4a*). These animals did not present with signs of liver toxicity measured as means of blood serum levels of liver proteins and indicated by the absence of alterations in histochemical analysis of liver sections (*Supplementary file 2*). To determine the general symptomatic stage, the animals were subjected to longitudinal weighting, and a battery of locomotor tests and phenotypic scoring, such as the total grading for inertia, gait, hindlimb clasping, tremor, irregular breathing and poor general conditions that together were presented as the aggregate severity score (*Guy et al., 2007*). *Mecp2*$^{-/y}$ mice treated with the median viral doses ($10^{10}$ vg and $10^{11}$ vg) maintained pronounced locomotor activity and exploratory behavior until few days before their sacrifice (*Figure 4—figure supplement 1*). In sharp contrast, administration of the lowest dose of virus ($10^{9}$ vg) was not sufficient to exert any detectable beneficial effect (*Figure 4—figure supplement 1*). Aforementioned immunofluorescence analysis showed a proportional relationship between the increasing doses of virus inoculated in the animals and the enhanced *Mecp2* gene transfer in the brain. With these data we can conclude that reversal of the *Mecp2*$^{-/y}$ pathological deficits is related both to transduction efficiency and protein levels achieved in the different districts of the brain. Indeed, when at least 70% of the brain cells were transduced in the cortex and striatum and protein levels were restored up to a physiological range, a sustained behavioral improvement was observed. Similar conclusions were drawn by *Robinson et al. (2012)* using reactivation of *Mecp2* endogenous gene expression in mice. In contrast brain transduction lower than 15% was not sufficient to attain detectable behavioral improvement.

Nevertheless, independently by the significant behavioral rescue, all the mice treated with $10^{11}$ vg particles were euthanized because of a severe tail necrosis unrelated to the disease phenotype (*Figure 4b*). This phenomenon presented only in $10^{11}$ vg treated mice (9 out of 12) and occurred 3–4 weeks after AAV injection. It started with a small lesion near the injection site, that progressively expanded to the entire tail and to the body showing a marked hair loss on the lumbar area. These mice were initially unaffected but while necrosis aggravated weight and general health started to be compromised and sacrifice was necessary. Overall none of the mice treated with lower doses of i*Mecp2* or GFP virus presented with this adverse symptom.

Given the systemic delivery of the virus in Mecp2 deficient mice, we set out to assess whether the immune response to the transgene could explain the health deterioration observed in the treated mice. This was unanticipated at least since none of the previous attempts of systemic gene therapy have considered or described such response to happen (*Gadalla et al., 2017*; *Garg et al., 2013*; *Sinnett et al., 2017*). However, Mecp2 has been implicated in the regulation of immunity (*Lal et al., 2009*) and of FoxP3 expression, a transcription factor required for the generation of regulatory T (Treg) cells, during inflammation (*Li et al., 2014*). Thus, we hypothesized that the lack of Treg-mediated regulation was responsible for the strong immune response observed in *Mecp2*$^{-/y}$ mice exposed to PHP.eB-iMecp2. Indeed, treatment with cyclosporine A (CsA) of *Mecp2*$^{-/y}$ mice exposed

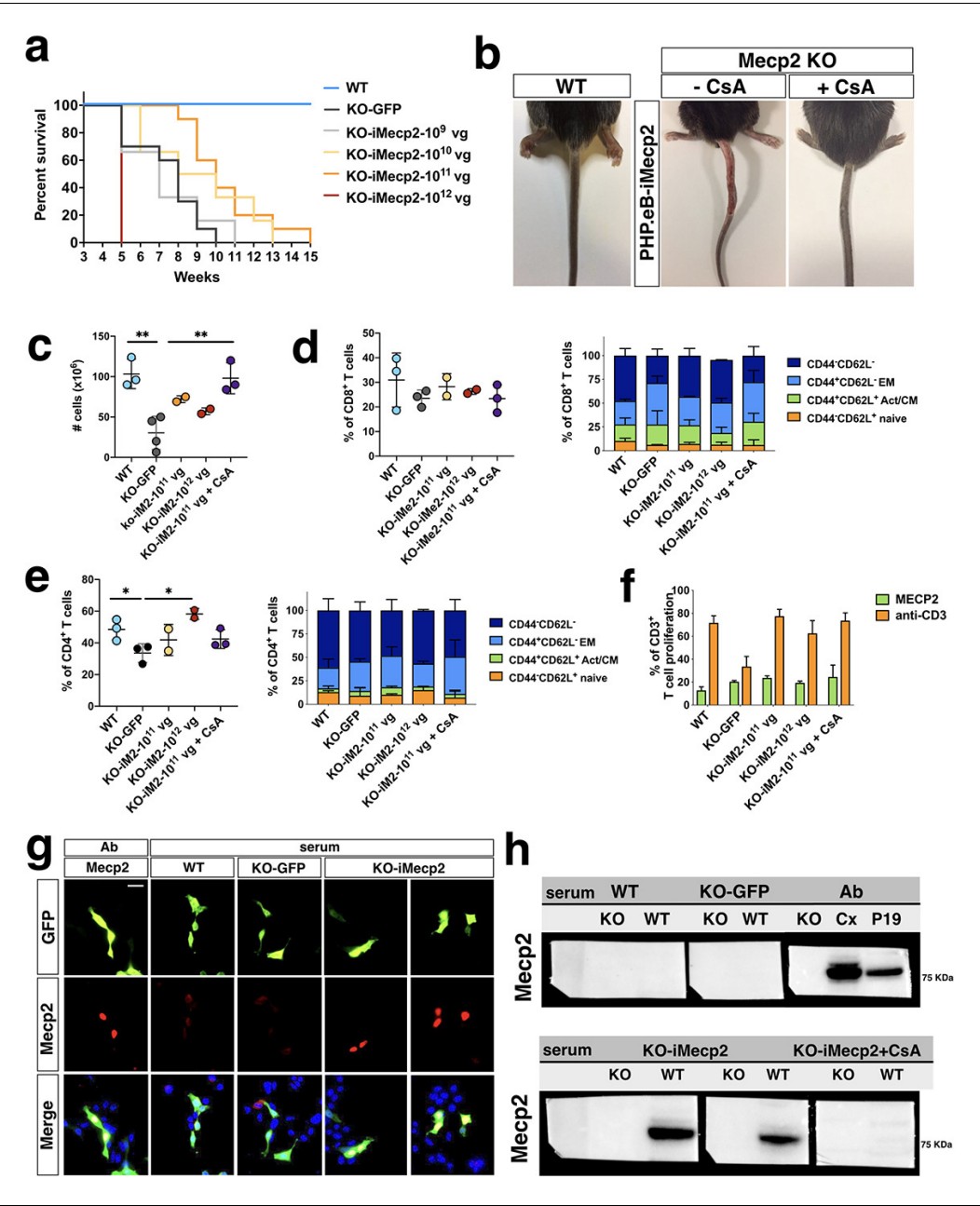

**Figure 4.** i*Mecp2* elicits a strong immune response in *Mecp2⁻/y* but not wild-type mice. (**a**) Kaplan-Meier survival plot for KO mice injected with different doses ($1\times10^9$ [n = 6], $1\times10^{10}$ [n = 6], $1\times10^{11}$ [n = 10], $1\times10^{12}$ [n = 6] vg/mouse) of PHP.eB-iMecp2 compared to *KO* treated with PHP.eB-GFP (KO-GFP, n = 10) and WT (n = 14) animals. Mice treated with $1\times10^{11}$ vg/mouse dosage had a median survival period significantly longer than that of vehicle-treated controls (p<0.01, Mantel-Cox test). (**b**) 9 out of 12 KO mice injected with a $1*10^{11}$ dose of PHP.eB-iMecp2 presented with exudative lesions after 2–3 weeks from viral injection (medial panel, representative picture), whereas 7 out of 10 KO mice injected with the same dosage and treated with cyclosporine (CsA) (10 mg/Kg) were robustly improved (right panel, representative picture). (**c–e**) Spleen cells from *Mecp2⁻/y* mice left untreated or injected with a $1\times10^{11}$ dose of PHP.eB-iMecp2 alone or in combination with CsA or with $1\times10^{12}$ dose of PHP.eB-iMecp2 were counted. Frequencies of CD4+ and CD8+ T cell compartments were quantified in the spleen of treated mice by FACS staining. (**f**) Total splenocytes were labeled with Cell Proliferation Dye eFluor 670 and stimulated with bone-marrow derived DC transduced with LV-*Mecp2* or with anti-CD3 antibodies and proliferation was measured at day 4 by flow cytometry. Mean ± SEM of *Mecp2⁻/y* mice untreated (n = 3) or injected with a $1\times10^{11}$ dose of PHP.eB-iMecp2 alone (n = 3) or in combination with cyclosporine (n = 3) or with $1\times10^{12}$ dose of PHP.eB-iMecp2 (n = 2) are shown. (**g**) Detection of immune-specific antibody in sera of KOanimals mock-treated

*Figure 4 continued on next page*

*Figure 4 continued*

(KO-GFP) or i*Mecp2*-treated (KO-i*Mecp2*) as well as i*Mecp2*-treated WT animals (WT) was revealed by immunofluorescence assay and compared with a commercial antibody as positive control (Ab). We choose as substrate P19 cells knock-out for the *Mecp2* gene and co-transfected with *GFP* and *Mecp2* expression constructs in order to track with GFP the Mecp2$^+$ cells. (h) Similar sera were also tested in western blot analysis using protein extracts form WT and KO tissue respectively as positive and negative controls. WT P19 extract were also used as positive control. Each dot represents one mouse. Error bars, SEM Scale bar: 10 μm. Mann-Whitney U test (two-tailed), with unpaired t-test (c–f), *p<0.05, **p<0.01, compared groups are indicated by black lines.

The online version of this article includes the following figure supplement(s) for figure 4:

**Figure supplement 1.** Behavioral rescue of symptomatic *Mecp2*$^{-/y}$ animals after PHP.eB-iMecp2 treatment.

to a 10$^{11}$ vg dose of PHP.eB-iMecp2 resulted in a striking amelioration of the general health conditions (*Figure 4b*). The analysis of the frequency of cells in the spleen of *Mecp2*$^{-/y}$ mice exposed to 10$^{11}$ vg and 10$^{12}$ vg of PHP.eB-iMecp2 (KO-iMecp2-10$^{11}$ vg and KO-iMecp2-10$^{12}$ vg, respectively) showed the increased number of splenocytes harvested from the latter mice compare to untreated *Mecp2*$^{-/y}$ controls (*Figure 4c,d*). Interestingly, *Mecp2*$^{-/y}$ mice showed a significantly lower number of splenocyte compared to WT littermates, in line with the general status of inflammation due to spontaneous activation of T cells in these mice (*Li et al., 2014*). No major differences were observed in CD8$^+$ T cell compartments in *Mecp2*$^{-/y}$ mice untreated or exposed to 10$^{11}$ vg and 10$^{12}$ vg of PHP.eB-iMecp2 virus (*Figure 4d*). Interestingly, *Mecp2*$^{-/y}$ mice exposed to the higher PHP.eB-iMecp2 dose (10$^{12}$ vg), analyzed two weeks post treatment, showed a significantly higher frequency of CD4$^+$ T cells compared to untreated control mice (*Figure 4e*). Hence, we can speculate that treatment with the higher dose of PHP.eB-iMecp2 virus in *Mecp2*$^{-/y}$ mice lacking Treg-mediated regulation led to an uncontrolled inflammatory response associated to the expansion of CD4$^+$ T cells that resulted in rapid death. We, next, investigated the induction of Mecp2-specific immune response by analyzing proliferation of T cells in response to Mecp2, cytotoxic activity of CD8$^+$ T cells, and anti-Mecp2 antibody production in *Mecp2*$^{-/y}$ mice exposed to PHP.eB-iMecp2 virus. Neither Mecp2-specific T cells nor Mecp2-specific cytotoxic CD8+ T cells were detected in *Mecp2*$^{-/y}$ mice exposed to PHP.eB-iMecp2 virus (*Figure 4f*, and data not shown). However, an increased non-specific (anti-CD3-stimulated cells) proliferative response was observed in *Mecp2*$^{-/y}$ mice exposed to PHP.eB-iMecp2 virus compared to untreated mice (*Figure 4f*). Finally, we detected anti-Mecp2 antibodies in the sera of *Mecp2*$^{-/y}$ mice exposed to PHP.eB-iMecp2 virus but not in WT mice exposed to PHP.eB-iMecp2 virus or in control untreated *Mecp2*$^{-/y}$ mice (*Figure 4g,h*). Importantly, treatment with CsA prevented the induction of anti-Mecp2 antibodies (*Figure 4h*). Overall, these studies indicate, for the first time, the induction of uncontrolled proliferation of T cells and induction of Mecp2-specific antibodies in *Mecp2*$^{-/y}$ mice exposed to PHP.eB-iMecp2 virus, which can be overcome by immunosuppression. This severe immune response to the transgene can explain the rapid phenotypic deterioration and premature death occurred to the *Mecp2*$^{-/y}$ mice treated with the highest dose of the therapeutic virus (10$^{12}$ vg). This conclusion is corroborated by the fact that WT and females *Mecp2*$^{+/-}$ animals exposed to the same virus at a comparable dose did not develop any of these complications (see below).

## PHP.eB-iMecp2 and Csa co-treatment triggers a significant behavioral rescue of male *Mecp2* mutant mice

Next, *Mecp2*$^{-/y}$ mice were co-treated with 10$^{11}$ vg PHP.eB-iMecp2 or PHP.eB-GFP with daily injections of Csa starting 1 day before viral administration and longitudinally profiled for survival rate, general healthy conditions and motor behavior. Remarkably, CsA-treated respect to untreated Mecp2 deficient mice administrated with PHP.eB-iMecp2 showed a triplicated mean survival rate, reaching up to 38 weeks of age (*Figure 5a*; median survival: KO-GFP: 52d; KO-GFP +Csa: 56d; KO-iMecp2-10$^{11}$: 68d; KO-iMecp2-10$^{11}$ + CsA: 168d). Unfortunately, not all iMecp2-treated animals responded to CsA. Uncomplete immunosuppression led to tail lesion and premature sacrifice in 3 out of 10 cases, nonetheless all mice presented significant behavioral improvement. In fact, this group of mice exhibited only very low grades (<3) in the aggregate severity score for an extensive period of time reflecting their general good healthy conditions (*Figure 5b*, *Video 1*). In particular, recovery in inertia, gait, hindlimb clasping, and irregular breathing were fully maintained over long

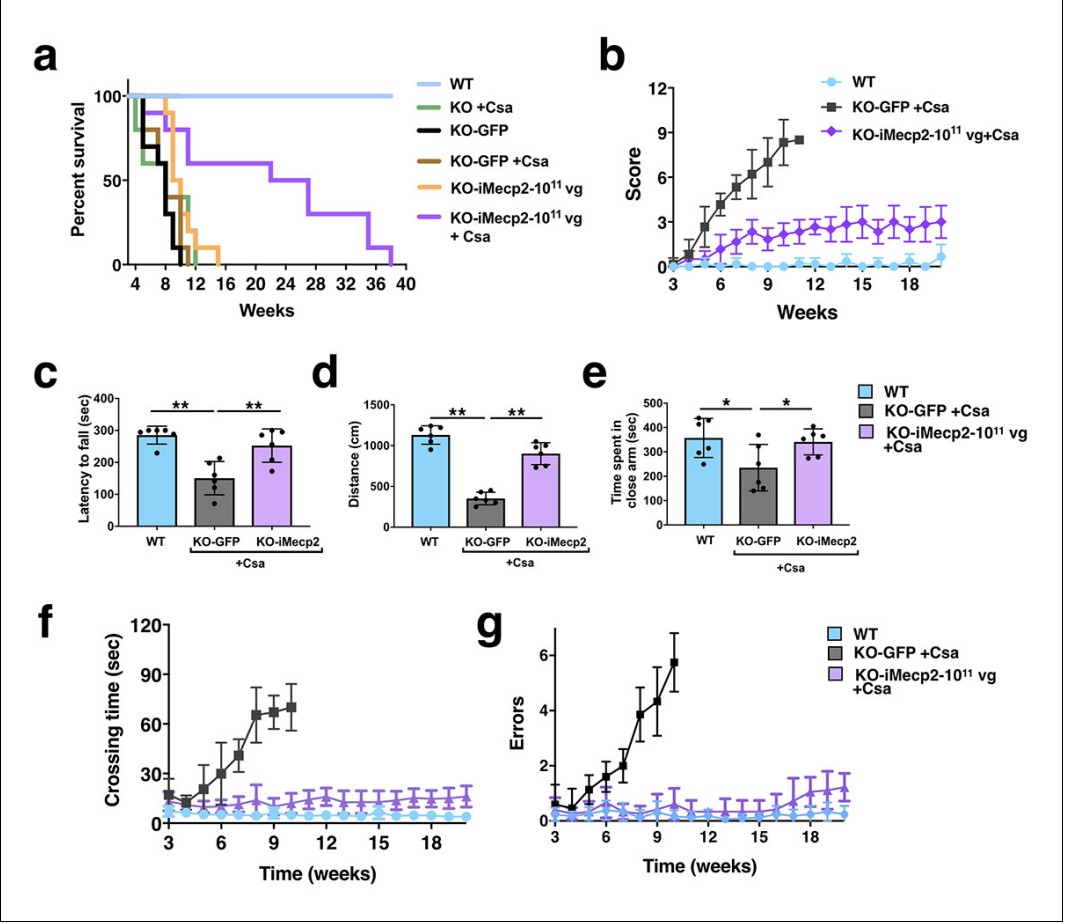

**Figure 5.** Behavioral rescue of symptomatic *Mecp2$^{-/y}$* animals after PHP.eB-iMecp2 treatment in combination with cyclosporine. (a) Survival curve of KO mice Cyclosporine (CsA) treated (n = 5, green line) and GFP-treated alone (n = 10, black line) or in combination with cyclosporine (CsA) (n = 6, brown line) or injected with a 1×10$^{11}$ dose of PHP.eB-iMecp2 alone (n = 10, orange line) or in combination with cyclosporine (CsA) (n = 10, violet line). As control, WT littermates were used (n = 12, blue line). (b) General phenotypic assessment evaluated through the aggregate severity score (p<0.05 versus KO-GFP + Csa in 1×10$^{11}$ + Csa [5$^{th}$-11$^{th}$ wk]). (c–e) Evaluation of the latency to fall from the Rotarod analysis (c), the distance of the spontaneous locomotor activity in a field arena (d) and the time spent in close arm in the elevated plus maze (e) 5 weeks after treatment and control mice (**p<0.01 and ***p<0.001 as compared to WT mice and KO-iMecp2 + Csa; n = 6 mice per groups). (f–g) Evaluation of motor coordination through beam balance test quantified as crossing time (f, p<0.05 versus KO-GFP + Csa in 1×10$^{11}$ + Csa [6$^{th}$-10$^{th}$ wk]) and number of errors (g, p<0.05 versus KO-GFP + Csa in 1×10$^{11}$ + Csa [7$^{th}$-10$^{th}$ wk]). Error bars, SD. ANOVA-two way (b, f, g) or ANOVA-one way (c, d, e) with Tukey's post hoc test.

time. Only some degree of tremors was still manifested in some of these mice accounting for the scores recorded in this test (*Figure 5b*). This manifestation is likely associated with abnormalities in the mutant peripheral nervous system which is known to be not efficiently transduced by this particular AAV capsid (*Chan et al., 2017*). Motor behavior was strongly rescued after gene therapy as shown by a significant recovery in the rotarod performance (*Figure 5c*) and spontaneous locomotor activity (*Figure 5d*) up to 15 weeks. *Mecp2$^{-/y}$* mice were previously reported to have reduced anxiety levels (*Santos et al., 2007*). In fact, control treated mutant mice tarried and moved less within the elevated closed arms, while PHP.eB-iMecp2 treated mice displayed anxiety levels more similar to WT animals (*Figure 5e*). Finally, sustained improvement in motor coordination and balance was also evident in the beam balance test (*Figure 5f,g*). In summary, this phenotyping assessment showed how a single injection of PHP.eB-iMecp2 adjuvated by the immunosuppressing treatment can elicit a robust and long-lasting recovery in survival (up to 38 weeks). More importantly the general health state and behavioral skills in *Mecp2$^{-/y}$* mice was also restored to levels comparable to wild-type

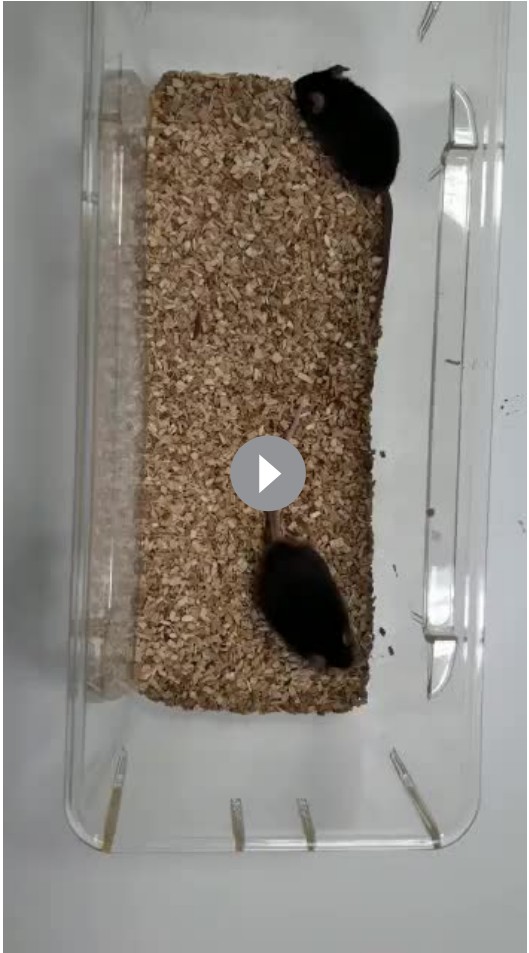

**Video 1.** Activity of *Mecp2*<sup>−/y</sup> treated with GFP or
$1\times10^{11}$ vg/mouse i*Mecp2* + cyclosporine. Both KO
mice were injected after 4 weeks from birth and were
video-recorded after 3 weeks (GFP, $1\times10^{11}$ vg/mouse)
or 4 months (i*Mecp2*, $1\times10^{11}$ vg/mouse) from the
treatment. The i*Mecp2*-treated mouse walks, climbs,
explores and interacts with the other mouse, while the
mock GFP-treated KO mouse remains stationary at one
corner of the cage. This video indicates an improved
mobility and an increased sociability after gene therapy
in KO mice.

https://elifesciences.org/articles/52629#video1

animals. This is the first clear evidence that a
gene therapy approach could rescue for a long
period RTT mice and significantly ameliorate their
general condition and motor behaviour. The
association of a significant life-span increment
and evident phenotypic amelioration is the main
goal of gene therapies applied to life treating
diseases.

## PHP.eB-iMecp2 reduces molecular and gene expression alterations in treated *Mecp2*<sup>-/y</sup> mice

In order to examine the molecular alterations
downstream to Mecp2 loss and evaluate whether
gene therapy with PHP.eB-iMecp2 might sustain
any discernable recovery, we performed global
gene expression analysis by RNA-Seq of whole
cerebral cortical tissue from 9 weeks old *Mecp2*<sup>-/y</sup> mice inoculated with either $10^{11}$ vg of PHP.eB-
iMecp2 or control virus and WT littermates.
Computational analysis identified 1876 differen-
tial expressed genes (DEGs) with p<0.05 signifi-
cance between *Mecp2*<sup>-/y</sup> and control mice
roughly divided in two equal groups between up-
and down-regulated genes in mutant mice
(*Figure 6a*). This dataset significantly overlaps
with the results published in a previous RNA-seq
study which profiled the same brain tissue at a
comparable mouse age, confirming the consis-
tency of our results (data not shown)
(*Pacheco et al., 2017*). The number of significant
DEGs between Mecp2<sup>-/y</sup> mice administrated with
therapeutic or control virus was 1271 with a small
increase in upregulated genes. However, only a
third of DEGs were shared between viral trans-
duced and untreated *Mecp2*<sup>-/y</sup> mice, while the
remaining DEGs of the mutant mice were normal-
ized in the treated counterparts (*Figure 6b,d,e*).
This data suggested that the viral treatment was
able to correct a large fraction of gene expres-
sion changes associated with the RTT phenotype.
However, a large set of DEGs was only associated
with the treated *Mecp2*<sup>-/y</sup> mice and, thus, to
uncover their significance we performed Gene
Ontology functional enrichment analysis (GO).

Remarkably, most of these DEGs were associated with immunological pathways such as immune
response, immune system regulation and inflammation (*Figure 6c*). Hence, these results corrobo-
rated at the molecular level the previous observations on the strong immune response mounted in
the treated *Mecp2*<sup>-/y</sup> mice against the inoculated transgene. We, then, performed gene ontology
analysis on the DEG dataset enriched in the *Mecp2*<sup>-/y</sup> mice but normalized after gene therapy. Inter-
estingly, the most significant enrichment was in metabolic networks associated with lipid biosynthe-
sis/transport and in particular cholesterol metabolism (*Figure 6f*). Previous studies have shown brain
and peripheral cholesterol levels are altered in *Mecp2* mutant mice and patients' specimens
(*Buchovecky et al., 2013*; *Park et al., 2014*; *Segatto et al., 2014*). In addition, gene responsible for
cholesterol biosynthesis has been shown to be downregulated in severe symptomatic *Mecp2*<sup>-/y</sup> ani-
mals (*Buchovecky et al., 2013*). Remarkably, a large component of the molecular pathway for

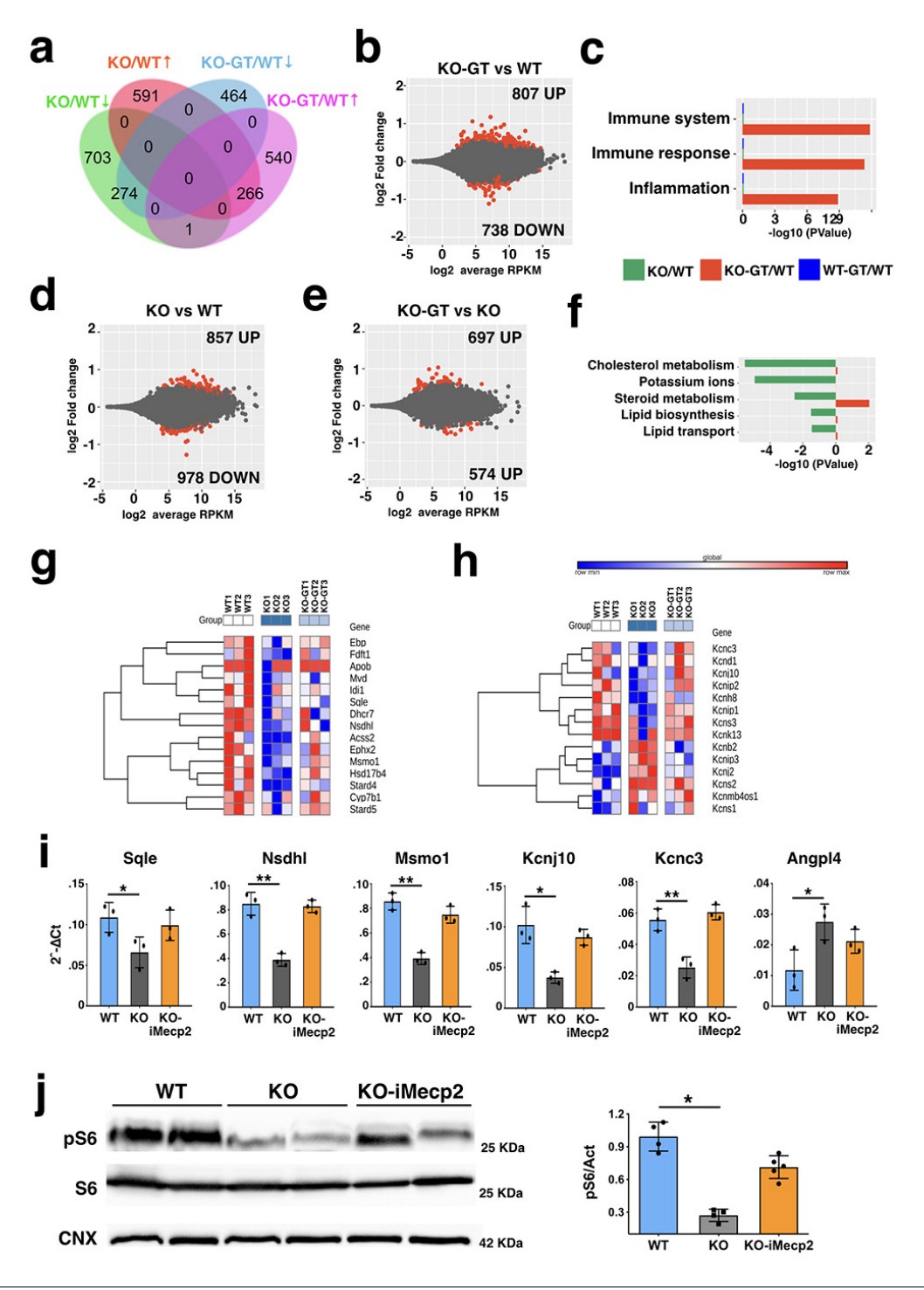

**Figure 6.** Global gene expression profile of *Mecp2*[-/y] cortices transduced with PHP.eB-iMecp2. (**a**) Venn diagram showing the genes differentially expressed in the two comparisons, namely *Mecp2* KO (supplied with a mock treatment with PHP.eB-GFP 10[11] vg/mouse, n = 3) vs WT (n = 3), and *Mecp2* KO-GT (after gene therapy treatment, PHP.eB-iMecp2 10[11] vg/mouse, n = 3) vs WT. (**b**) MA plot showing gene expression fold changes as a function of the average gene expression in the *Mecp2* KO-GT vs WT comparison. (**c**) Representative gene ontology categories highlighting the enrichment for immune response-related datasets being overrepresented in the *Mecp2* KO-GT vs WT comparison. (**d**) MA plot showing gene expression fold changes as a function of the average gene expression in the *Mecp2* KO vs WT comparison. (**e**) MA plot showing gene expression fold changes as a

*Figure 6 continued on next page*

*Figure 6 continued*

function of the average gene expression in the *Mecp2* KO-GT vs KO comparison. (**f**) Representative gene ontology categories highlighting the enrichment for lipid metabolism-related datasets being overrepresented in the *Mecp2* KO vs WT comparison, but not in the *Mecp2* KO-GT vs KO comparison. Heatmap showing relative expression of genes belonging to the lipid metabolism-related pathways (**g**) or Potassium ion transmembrane transport group (**h**). (**i**) RT-qPCRs of selected transcript of interest such as *Sqle*, *Nsdhl*, *Msmo1*, *Kcnj10*, *Kcnc3* and *Angpl4* being downregulated in *Mecp2* KO and rescued after gene therapy. Red dots in (**b**), (**d**), and (**e**) depict differentially expressed genes with FDR $\leq$ 0.1. (**j**) Representative western blot and quantitative analysis for the ribosomal protein S6, its phosphorylated form (pS6) and a normalizer calnexin (CNX) from cortical tissues of wild-mice and *Mecp2* KO transduced with GFP or iMecp2 vector. Error bars, SD. *$p<0.05$, **$p<0.01$ as compared to WT mice (ANOVA-one way with Tukey's post hoc test).

cholesterol production was downregulated in $Mecp2^{-/y}$ mice, but significantly rescued in PHP.eB-iMecp2 transduced animals (*Figure 6g*). RT-qPCRs on independent cortical tissue lysates confirmed that gene expression levels of crucial enzymes in the cholesterol biosynthesis such as Squalene epoxidase (*Sqle*), NAD(P)-dependent steroid deydrogenease-like (*Nsdhl*) and Methylsterol monooxygenase 1 (*Msmo1*) were significantly restored by PHP.eB-iMecp2 gene therapy (*Figure 6i*). An additional molecular group highly divergent between transduced and control $Mecp2^{-/y}$ mice included genes encoding for potassium (Kv) channels. This class of ion channels serve diverse functions including regulating neuronal excitability, action potential waveform, cell volume and fluid and pH balance regulation. Interestingly, it was previously reported that *Mecp2* deficiency leads to decreased *Kcnj10/Kir4.1* mRNA levels and related currents in mutant astrocytes (*Kahanovitch et al., 2018*). We confirmed reduced *Kcnj10* transcripts together with gene deregulation of other Kv channels in $Mecp2^{-/y}$ mice (*Figure 6h,i*). Among others, the potassium channel gene *Kcnc3*, associated with ataxia and cognitive delay in humans, was downregulated in *Mecp2* mutants and normalized after the PHP.eB-iMecp2 treatment (*Figure 6h,i*). We previously showed that *Mecp2* mutant brains exhibit reduced mTOR signaling with diminished phosphorylation of phospho-S6 (pS6) (*Ricciardi et al., 2011*). Remarkably, overall levels of pS6 on Ser234/235 were significantly increased in brain tissue transduced with the PHP.eB-iMecp2 virus (*Figure 6j*). In summary, PHP.eB-iMecp2 gene therapy sustained a wide recovery of the abnormal gene expression in the *Mecp2* mutant brain tissue and elicited the rescue of the global impairment affecting transcriptional and translational processes upon *Mecp2* gene loss.

## PHP.eB-iMecp2 improves disease symptoms in female *Mecp2* heterozygous mice

$Mecp2^{-/y}$ mice recapitulate the severe neuronal deficits exhibited by RTT patients but do not model the mosaic gene inactivation occurring in girls with RTT. Moreover, the strong immune response has complicated the readout of the symptomatic recovery of gene therapy in these animals. Thus, we thought to validate our approach in female $Mecp2^{+/-}$ mutant mice (Het). To this end, 5 months old $Mecp2^{+/-}$ animals were intravenously injected with $10^{11}$ vg of either PHP.eB-iMecp2 or control virus and examined over time up to 11 months post-treatment (*Figure 7a*). For these experiments, we selected only a single viral dose that sustained a general brain transduction rate between 65% and 90% and raising Mecp2 protein levels similar to those in WT (*Figure 7g,h,I,j*). As previously reported, $Mecp2^{+/-}$ females started to exhibit pathological signs from 10 months of age acquiring breathing irregularities, ungroomed coat, inertia and hindlimb clasping (*Guy et al., 2001*). While control treated mutant females showed a pronounced and sustained weight gain over time, the animals with the viral therapy gradually normalized their weight reaching values similar to those of unaffected mice at 15 months (*Figure 7b*). In the severity score the control treated $Mecp2^{+/-}$ females progressed to values over 4, whereas animals given the therapeutic virus rarely overcome a score beyond 2 in the entire observation period (*Figure 7c*). Total mobility assessment in the open field showed that at 13 months old PHP.eB-iMecp2 treated mice have a significant increase in the travelled distance and general activity respect to the control treated group matching the general performance of WT females (*Figure 7d*). Likewise, in the beam balance test the therapy largely rescue the motor skills of the mutant mice (*Figure 7e,f*). After 11 months from the injection, mice were euthanized to analyze the transgene expression levels and the transduction profile. Intriguingly, we

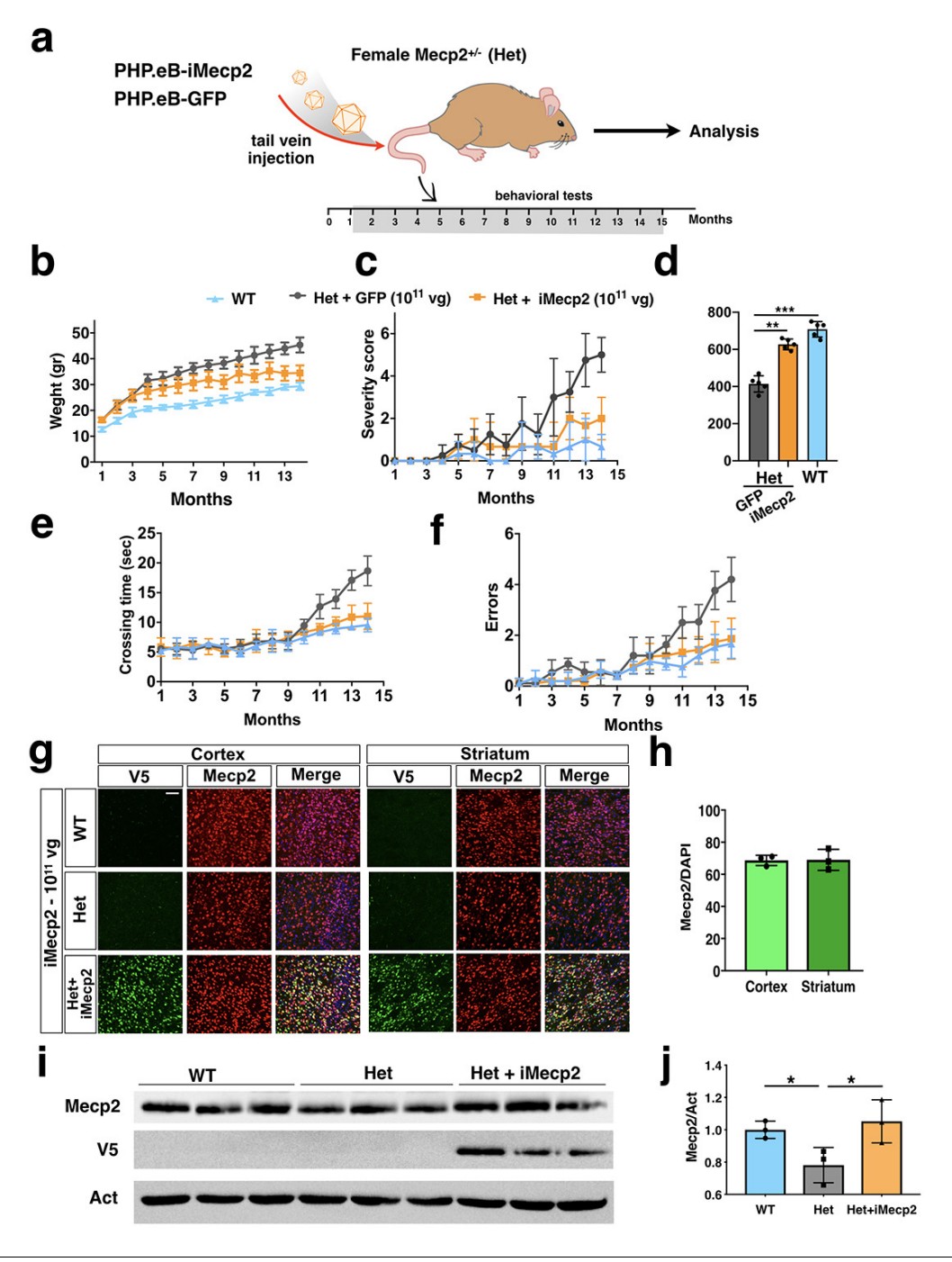

**Figure 7.** PHP.eB-iMecp2 gene transfer in *Mecp2*⁺ᐟ⁻ females. (**a**) Illustration of the experimental setting to restore the expression of Mecp2 in heterozygous (Het) animals by means of AAV-PHP.eB. (**b**) Mouse body weight was monitored every two weeks and represented as mean of each group. (**c**) General phenotypic assessment evaluated through aggregate severity score. (**d**) Spontaneous locomotor activity was tested in a spontaneous field arena and shown as quantification of travelled total distance. All groups of animals were tested for motor coordination using beam balance test and quantified as crossing time (**e**) and number of errors (**f**) (WT untreated [n = 8], Het treated with GFP $1\times10^{11}$ vg/mouse [n = 6], Het treated with i*Mecp2* virus $1\times10^{11}$ vg/mouse [n = 6]). (**g**) High magnification immunostaining for Mecp2 and V5 in cortex, striatum and cerebellum derived from WT, *Mecp2*⁺ᐟ⁻ (Het) untreated and Het treated with PHP.eB-iMecp2 ($1\times10^{11}$ vg/mouse) brains. Nuclei were stained with DAPI (merge panels). (**h**) Bar graphs showing the fraction of V5 positive on the total DAPI positive in cortex and striatum (n = 3). (**i**) Western-blot analysis to quantify Mecp2 over Actin protein levels in cortex derived from WT,
*Figure 7 continued on next page*

*Figure 7 continued*

Het untreated and treated with i*Mecp2* ($1\times10^{11}$) animals and (j) corresponding densitometric quantification expressed in arbitrary units (right panel) (n = 3 for WT, n = 3 Het untreated; n = 3 Het treated with PHP.eB-iMecp2 $10^{11}$ vg/mouse) *p<0.001 as compare to WT and Het treated with PHP.eB-iMecp2 (ANOVA-one way, Tukey's post hoc test). Error bars, SD. **p<0.01. ANOVA-two way (b, c, e, f) or ANOVA-one way (d) with Tukey's post hoc test. The online version of this article includes the following figure supplement(s) for figure 7:

**Figure supplement 1.** Characterization of i*Mecp2* transgene expression in brain and liver of *Mecp2*$^{+/-}$ animals.

observed that the levels of viral genome copies and transgene expression were higher in cortex in comparison with liver (*Figure 7—figure supplement 1a*), in accordance with AAV genome clearance caused by hepatocellular proliferation (*Dane et al., 2009*). In line with this result, the transduction profile of neurons and astrocytes was comparable to that observed in *Mecp2*$^{-/y}$ mice treated with the same dose (*Figure 7—figure supplement 1b,c*). Collectively, these observations provide evidence that the PHP.eB-iMecp2 treatment sustained a significant and long-term protection from symptomatic deterioration improving the health conditions and reducing the locomotor phenotype in female *Mecp2*$^{+/-}$ mice.

## Systemic gene transfer of i*Mecp2* is not detrimental for wild-type mice

Systemic delivery of PHP.eB-iMecp2 exerted a large symptomatic reversibility both in male and female Mecp2 mutant mice. To further extend these observations and validate the safety of this treatment we decided to administer the same treatment to WT C57BL/6J adult mice. Animals were administrated with either $10^{11}$ vg or $10^{12}$ vg of PHP.eB-iMecp2 or left untreated (n = 9 each) and closely inspected over time. Next, two animals per group were euthanized 3 weeks after viral inoculation and brain processed for histological analysis. i*Mecp2* gene transfer efficiency was evaluated by V5 immunofluorescence which distinguished the viral from the endogenous *Mecp2*. According to aforementioned results, brain transduction efficiency was very high with a net increase of 20% between the lower and the higher viral dose (cortex: 45 ± 8% $10^{11}$ vg, 68 ± 7% $10^{12}$ vg; striatum: 58 ± 7%, $10^{11}$ vg; 82 ± 5%, $10^{12}$ vg) (*Figure 8a,b*). Viral copy number analysis confirmed a significant and prevalent targeting of the virus in the neural tissues respect to the liver (*Figure 8—figure supplement 1*). Despite the high expression of the transgene, the total Mecp2 protein levels in cortex and striatum were only increased by 30% and 60% upon transduction with $10^{11}$ vg and $10^{12}$ vg of virus, respectively as assessed by immunoblotting and immunofluorescence intensity (*Figure 8c*, *Figure 8—figure supplement 2*). General health state and behavior were, then, scored in the remaining treated animals up to 12 weeks after treatment. During this time, growth curve, locomotor activity and fine coordination were slightly different between transduced and control animals (*Figure 8d,e*). General health state examination (severity score) revealed some breathing irregularities and tremor at rest which slightly increased the scoring output in the treated animals although with minimal difference compared to WT animals (*Figure 8f*). Altogether, these observations indicated that high doses of PHP.eB-iMecp2 virus did not exert deleterious effects in WT animals in this window of time. Importantly, even the $10^{12}$ viral dose, which is 10-fold higher than the amount used in *Mecp2*$^{-/y}$ mice to trigger substantial beneficial effects, was incapable to trigger a consistent deleterious outcome in the mice except for mild alterations. Next, we performed global gene-expression analysis by RNA-seq from cerebral cortical tissues of animals untreated or inoculated with a $10^{11}$ vg dose. Remarkably, bioinformatics analysis did not distinguish genes with significantly different expression between the two conditions (p<0.05) (*Figure 8g,h*). Collectively, widespread brain transduction of PHP.eB-iMecp2 in WT animals elicited only a minimal increase in total Mecp2 levels which was not sufficient to exert neither significant behavioral symptoms nor abnormal gene expression changes.

## Discussion

Herein, we provided solid evidence that the global brain transduction of the i*Mecp2* transgene by PHP.eB-mediated delivery was capable to significantly protect male and female *Mecp2* mutant mice from the symptomatic hallmarks of the RTT phenotype. Importantly, an initial set of viral infections on primary neuronal cultures enabled us to select the most suitable configuration of the transgene

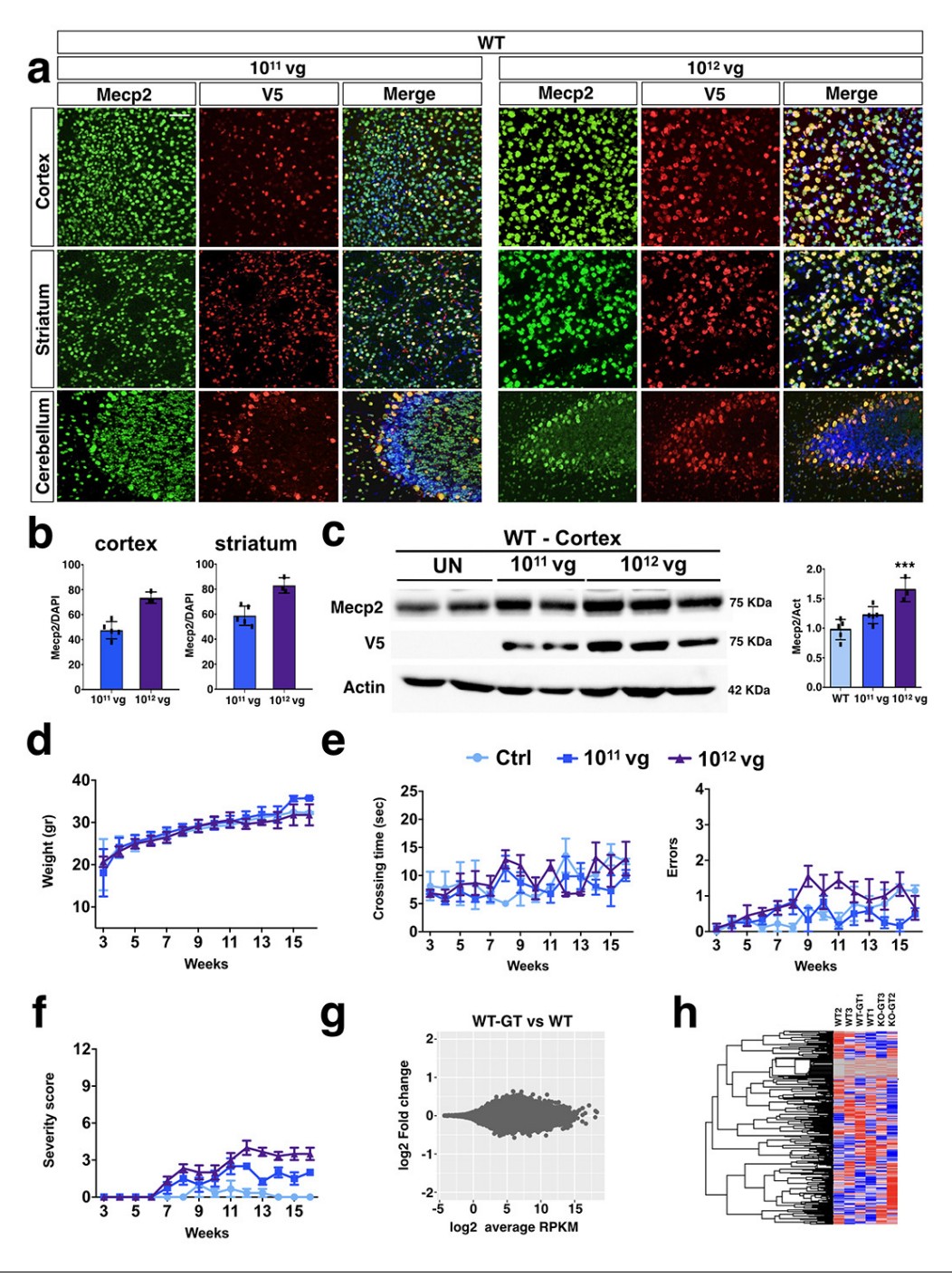

**Figure 8.** PHP.eB-mediated i*Mecp2* gene transfer in wild-type mice is unharmful. (**a**) High magnification immunostaining for Mecp2 and V5 in cortex, striatum and cerebellum derived from WT brains treated with PHP.eB-iMecp2 ($1 \times 10^{11}$ and $1 \times 10^{12}$ vg/mouse). Nuclei were stained with DAPI (merge panels). (**b**) Bar graphs showing the fraction of V5 positive on the total DAPI positive in cortex and striatum (n = 6 for $1 \times 10^{11}$ and n = 3 for $1 \times 10^{12}$ vg/mouse). (**c**) Western blot analysis to quantify Mecp2 over Actin protein levels in cortex (left panel) and striatum (right panel) derived from WT, untreated and treated with i*Mecp2* ($1 \times 10^{11}$ and $1 \times 10^{12}$ vg/mouse) animals and corresponding densitometric quantification expressed in arbitrary units (right panel) (n = 6 for WT, n = 6 for $10^{11}$ and n = 3 for $10^{12}$ vg/mouse) ***p<0.001 as compare to wild-type (WT) untransduced mice (ANOVA-one way, Tukey's post hoc test). Twice a week, mice were tested for (**d**) body weight, (**e**) beam balance test and quantified as crossing time (left) and number of errors (right) and (**f**) general phenotypic assessment evaluated through aggregate severity score (WT untreated [n = 8] and treated with i*Mecp2* virus $1 \times 10^{11}$ [n = 6], $1 \times 10^{12}$ [n = 6], vg/

*Figure 8 continued on next page*

*Figure 8 continued*

mouse). (**g**) MA plot showing gene expression fold changes as a function of the average gene expression in the *Mecp2* WT-GT (after gene therapy treatment, PHP.eB-iMecp2 $10^{11}$ vg/mouse, n = 3) vs WT (n = 3) comparison (**h**) Red dots depict differentially expressed genes with FDR $\leq$ 0.1. Heatmap showing relative expression of 1000 random genes in WT and WT-GT, highlighting the lack of differentially expressed genes in the two groups. Error bars, SD. Scale bar: 50 μm.
The online version of this article includes the following figure supplement(s) for figure 8:

**Figure supplement 1.** Characterization of i*Mecp2* transgene expression in brain and liver of wild-type animals.
**Figure supplement 2.** Protein levels in PHP.eB-iMecp2 transduced wild-type animals.

cassette in order to achieve Mecp2 protein levels within a physiological range. Our results showed that the lack of a long 3'-UTR sequence destabilizes *Mecp2* mRNA significantly reducing its relative half-time. Moreover, all the combinations of *Mecp2* regulatory regions tested in this work did not enhance the i*Mecp2* transgene RNA stability. Thus, more work is necessary to identify additional *Mecp2* transcript stabilizing elements in the long 3'UTR. Additionally, using the RiboLace system to determine the amount of *Mecp2* transcripts associated with active translating ribosomes, our data showed a major loss in translation efficiency of the viral compared to the endogenous mRNA. These results strongly imply the strong functional impact of untranslated elements in the post-transcriptional control of Mecp2 expression. Along these lines, it was previously reported that endogenous Mecp2 protein levels correlate poorly with mRNA levels during development and in different adult tissues (*Shahbazian et al., 2002*). For this reason, we favored to combine the *Mecp2* cDNA with the CBA strong promoter, building the i*Mecp2* cassette, in order to reach physiological range of Mecp2 proteins levels. Based on our in vitro results we be believe that more broadly, a preliminary in vitro screening in primary neuronal cultures is a valuable tool to characterize the properties of newly designed transgene cassette and vector transduction.

We decided to package the i*Mecp2* construct in the PHP.eB that was selected for its unprecedented capability to efficiently penetrate the brain after intravenous delivery and widely transduce neuronal and glial cells (*Chan et al., 2017*). We showed that administration of high dose of PHP.eB-iMecp2 at the initial symptomatic stage can ameliorate disease progression with robust beneficial effects and significant lifespan extension. Moreover, we observed that both transduction efficiency and Mecp2 protein levels can impact the final outcome of phenotypic recovery. Indeed, animals treated with $10^9$ vg dose of PHP.eB-iMecp2 (which transduced less than 15% of brain cells cortex and striatum) did not present significant therapeutic effects. This is only apparently different from previous observations reporting a phenotypic improvement in mutant mice with restoration of Mecp2 in only 5–10% of total brain cells (*Carrette et al., 2018*). In fact, in our study, low-titer viral transduction targeted much better astrocytes rather than neurons, revealing a higher tropism of this viral capsid for the former cells. Thus, weak neuronal transduction (<5% of total neurons) could be the reason for the absence of any phenotypic improvement. An additional difference is that in the present work we reconstituted Mecp2 function only in adulthood, differently from Mecp2 gene reconstitution through mouse breeding which occurs already during embryonic development. Thus, the phenotypic outcome of the gene-transfer protocols is highly dependent by the viral dose and the specific tropism of the capsid and not always is directly comparable to genetic rescue achieved with different approaches.

Notwithstanding, the intermediate viral doses ($10^{10}$ e $10^{11}$) had respectively moderate and robust effect in reverting phenotypic symptoms. In this last case the beneficial effects of the treatment were concealed by an adverse effect consisting of a strong immune response against the transgene. Importantly, this adverse complication is the result of using the male *Mecp2* KO mice that have no Mecp2 from birth and therefore recognize the therapeutic gene product as *nonself*. This is not the case for human female patients that are a mosaic of mutant and WT *Mecp2* cells, moreover even male hemizygous patient rarely present with protein complete loss, thus immune response against this transgene is unlikely to occur in human patients. Nonetheless this is a clinical aspect that need to be carefully considered during patient selection. It was initially surprising to us why former studies of *Mecp2* gene therapy in KO mouse strains did not highlight such adverse event (*Gadalla et al., 2017*; *Garg et al., 2013*). A possible explanation could be related to the different tropism of the PHP.eB in comparison to AAV9.

The life-span of animals treated with $10^{12}$ vg/mouse dose was not long enough to justify a similar response thus indicating a different mechanism of toxicity. This serious inflammatory response to the transgene, associated to the expansion of CD4$^+$ T cells, can explain the sudden death of the *Mecp2$^{-/y}$* animals treated with the highest PHP.eB-iMecp2 dose ($10^{12}$ vg). In fact, a similar dose of the therapeutic virus in WT animals was free of deleterious effects. Despite previous reports (*Gadalla et al., 2017*), we could not detect any sign of toxicity in livers treated with PHP.eB, irrespectively of the transgene or the mice genotype, thus ruling out epatotoxicity in these conditions.

Beyond this immune reaction, we could not score any additional adverse manifestations directly caused by the therapy neither in mutant *Mecp2* or WT animals. This is consistent with the mild increase in total Mecp2 levels achieved by the i*Mecp2* transgene both in a mutant or WT background. Nonetheless we cannot exclude that the deleterious effect observed in mice treated with the highest dose ($10^{12}$ vg/mouse) could be related to an increase of Mecp2 above physiological level (~3 fold). In fact, abrupt *Mecp2* reactivation in Mecp2 knock-out mice has been reported to often lethal (*Guy et al., 2007*).

Nevertheless, viral ($10^{11}$ vg) and CsA co-treatment was able to promote an unprecedented life-span recovery and robust improvement in general health state, locomotor activity and coordination. After months of cyclosporine treatment, the mice showed a deterioration in quality life and eventually had to be euthanized mainly for symptoms related to immunosuppression or relapsing immune response. PHP.eB-iMecp2 intervention significantly corrected also the abnormal gene expression alterations observed in *Mecp2$^{-/y}$* mouse cortical tissue, including the reduced expression in multiple molecular components of the cholesterol pathway and some genes encoding for Kv channels. More broadly, PHP.eB-iMecp2 gene therapy sustained a strong recovery of the genome-wide transcriptional and mTOR-mediated translational processes affected in *Mecp2* deficient mice.

The molecular characterization *Mecp2$^{-/y}$* mice rescue combined with its phenotypic analysis is a powerful tool to assess molecular pathways involved in the disease and more importantly in its recovery.

Taken together, the diffuse penetration of the PHP.eB in the adult mouse brain parenchyma is a unique property among all the recombinant viral strains in current use. Recently, two studies concomitantly discovered Ly6a as the endothelial receptor that allows PHP.B and PHP.eB variants to cross the BBB and diffuse the brain parenchyma of mice (*Hordeaux et al., 2019*; *Huang et al., 2019*). Interestingly, this receptor is not conserved in primates and humans, thus implying species-specific difference in the transduction efficacy of these capsids. On this line, new findings indicate that brain transduction in adult primates is not improved with PHP.B compared to AAV9 although only few animals have yet been tested (*Hordeaux et al., 2018*; *Matsuzaki et al., 2017*). Nonetheless, the PHP.B/eB receptor belongs to the large family of Ly6/uPar proteins, some of which are conserved in mammalian evolution and can be found in human brain endothelium, representing valuable targets for capsid engineering (*Loughner et al., 2016*; *Kong et al., 2012*). In this prospective, the PHP.B platform is fundamental to test the validity of new gene therapy strategies in mice models that can be translated into the clinical setting as soon as, in the next future, PHP.B-like AAVs penetrating the primate brain will be confirmed. Despite this caveat, RTT is a neurodevelopmental disorder and the therapeutic intervention should be finalized in the first months after birth as soon as the early signs of the disease manifest and genetic diagnosis is certain. At similar age, an intravenous infusion of AAV9 particles packaging the *SMN1* gene resulted in extended survival and improved motor functions in infants suffering for spinal muscular atrophy (*Mendell et al., 2017*). Thus, it is plausible that at this early age, AAV9 systemic gene therapy might sustain a beneficial clinical outcome in RTT patients and even more so using AAV9 synthetic variants selected in the near future. Altogether in this study we characterized an i*Mecp2* viral cassette which sustained significant improvements in transgene delivery, safety and efficacy leading to the long-term symptomatic amelioration of RTT mice. These results might accelerate the introduction of new gene therapy strategies for RTT with clinical prospective.

## Materials and methods

### Animals

Mice were maintained at San Raffaele Scientific Institute Institutional mouse facility (Milan, Italy) in micro-isolators under sterile conditions and supplied with autoclaved food and water. The *Mecp2⁻/y* mice (*Guy et al., 2001*) (The Jackson Laboratory stock *#003890*, RRID:MGI:3624717) were a kind gift of N. Landsberger and were maintained on C57BL/6J background. All procedures were performed according to protocols approved by the internal IACUC and reported to the Italian Ministry of Health according to the European Communities Council Directive 2010/63/EU.

### Generation of gene transfer vectors

All the Mecp2 and control AAV vectors were cloned into single stranded constructs. The murine *Mecp2* isoform 1 (NM_001081979.2) CDS including 3'-UTR (223 bp) was PCR amplified in order to add the V5 tag at the 5' of the coding sequence and inserted in the CBA-CreNLS vector to generate the M2a i*Mecp2* vector (*Figure 1a*; *Morabito et al., 2017*). The CBA promoter consists of 3 independent elements: a CMV enhancer, the chicken β-actin promoter and the human β-globin first intron. The CBA promoter was removed from the CBA-V5-*Mecp2* vector to be replaced by the mouse *Mecp2* promoter region (1,4 Kb) including the 5'-UTR to generate the M2b vector (*Figure 1a*). The AAV-CBA-V5-GFP construct was engineered from AAV-CBA-V5-Mecp2 vectors by exciding *Mecp2* CDS and exchanging with a GFP cassette. The 3'-UTR of the M2a construct was replaced by an assembled 3'-UTR (aUTR, 223 bp, including portions of the 8,6 Kb murine *Mecp2* endogenous 3'-UTR, such as: miRNA22-3p, 19–3 p and 132–3 p target sequence and the distal polyA signal of *Mecp2* gene) (*Gadalla et al., 2017*) created by gene synthesis (Genewiz) or removed to generate respectively the M2c and M2e construct (*Figure 1a*), whereas the M2-d construct was made from M2-a vector by insertion of the murine *Mecp2* 5'UTR upstream of the V5 sequence (*Figure 1a*). In order to generate the iMECP2 vector the murine Mecp2 CDS and 3'-UTR of M2a was replaced by the human *MECP2* isoform_2 (NM_001110792.2) CDS including the human 3'-UTR (225 bp) of *MECP2* gene. This isoform was chosen since is the human orthologue of murine *Mecp2* isoform_1. Both *NGFR* (Nerve Growth Factor Receptor) and *Mecp2* (comprehensive of 3'-UTR) amplicons were digested and cloned into a lentiviral vector (LV-Ef1a-GFP) in which the GFP cassette was removed.

### Virus production and purification

Lentiviral replication-incompetent, VSVg-coated lentiviral particles were packaged in 293 T cells (*Vierbuchen et al., 2010*). Cells were transfected with 30 µg of vector and packaging constructs, according to a conventional $CaCl_2$ transfection protocol. After 30 hr, medium was collected, filtered through 0.44 µm cellulose acetate and centrifuged at 20000 rpm for 2 hr at 20°C in order to concentrate the virus.

AAV replication-incompetent, recombinant viral particles were produced 293 T cells, cultured in Dulbecco Modified Eagle Medium – high glucose (Sigma-Aldrich) containing 10% fetal bovine serum (Sigma-Aldrich), 1% non-essential amino acids (Gibco), 1% sodium pyruvate (Sigma-Aldrich), 1% glutamine (Sigma-Aldrich) and 1% penicillin/streptomycin (Sigma-Aldrich). Cells were split every 3–4 days using Trypsin 0.25% (Sigma-Aldrich). Replication-incompetent, recombinant viral particles were produced in 293 T cells by polyethylenimine (PEI) (Polyscience) co-transfection of three different plasmids: transgene-containing plasmid, packaging plasmid for rep and cap genes and pHelper (Agilent) for the three adenoviral helper genes. The cells and supernatant were harvested at 120 hr. Cells were lysed in hypertonic buffer (40 mM Tris, 500 mM NaCl, 2 mM $MgCl_2$, pH = 8) containing 100 U/ml Salt Active Nuclease (SAN, Arcticzymes) for 1 hr at 37°C, whereas the viral particles present in the supernatant were concentrated by precipitation with 8% PEG8000 (Polyethylene glycol 8000, Sigma-Aldrich) and then added to supernatant for an additional incubation of 30 min at 37°C. In order to clarify the lysate cellular debris where separated by centrifugation (4000 g, 30 min). The viral phase was isolated by iodixanol step gradient (15%, 25%, 40%, 60% Optiprep, Sigma-Aldrich) in the 40% fraction and concentrated in PBS (Phosphate Buffer Saline) with 100K cut-off concentrator (Amicon Ultra15, MERCK-Millipore). Virus titers were determined using AAVpro Titration Kit Ver2 (TaKaRa).

## Primary mouse neuronal cultures

Primary Neuronal culture were prepared at embryonic day 17.5 (E17.5) from male mouse embryos. Briefly, cortices were individually dissected, sequentially incubated in trypsin (0,005%, 20 min at 37° C, Sigma-Aldrich) and DNAse (0,1 mg/mL, 5 min at room temperature, Sigma-Aldrich) in HBSS (Hank's buffered salt solution without $Ca^{2+}$ and $Mg^{2+}$, Euroclone). Cells were finally and plated on poly-L-lysine (Sigma-Aldrich) coated dishes ($2.0 \times 10^5$ cells/cm$^2$) in Neurobasal medium (ThermoFisher Scientific) enriched with 0,6% glucose (Sigma-Aldrich), 0,2%penicillin/streptomycin (Sigma-Aldrich), 0,25% L-glutamine (Sigma-Aldrich) and 1% B27 (ThermoFisher Scientific). Viral particles were directly added to cultured neurons 3 days after seeding, with a final concentration $10^{10}$ vg/ml.

## AAV-PHP.eB vector injection, mouse phenotyping and tissue collection

Vascular injection was performed in a restrainer that positioned the tail in a heated groove. The tail was swabbed with alcohol and then injected intravenously with a variable viral concentration (from $1*10^{10}$ to $1*10^{13}$ vg/mL) depending on the experimental setup in a total volume of 100 µl of AAV-PHP.eB particles in PBS.

Juvenile WT *and Mecp2$^{-/y}$* mice were randomized in groups and injected in the tail vein between 25 and 30 days of age. Adult WT were injected in a similar time window whereas *Mecp2$^{+/-}$* females were treated intravenously after five months of life. Following injection, all mice were weighed twice a week. Phenotyping was carried out, blind to genotype and treatment, twice a week. Mice symptoms were scored on an aggregate severity scale (0 = absent; 1 = present; 2 = severe) comprising mobility, gait, breathing, hindlimb clasping, tremor, and general condition. The balance and the motor coordination were assessed by the Beam Balance test. Briefly mice were placed on the tip of the beam at the 'start-point' facing towards the beam. The number of foot-slips (error numbers) and total time on beam (crossing time) from 'start' to 'end' points were noted. If a mouse fell, the animal was returned to the site where it fell from, until completion of beam crossing.

The Open Field test was performed in an arena of $50 \times 50$ cm. Mice were testing in a 10 min session measured as an index of anxiety and horizontal exploratory activity in a novel environment was assessed.

For serum analysis blood samples were collected from living animals using retro-orbital bleeding procedure with non-heparinized capillaries. Upon blood clothing cell fraction was pelleted (5 min, 13000 rpm) and supernatant recovered. When the body loss reached 20% of total weight mice were sacrificed and tissues harvested. Briefly, mice were anesthetized with ketamine/xylazine and transcardially perfused with 0.1 M phosphate buffer (PB) at room temperature (RT) at pH 7.4. Upon this treatment brain, liver and spleen were collected. Brain hemispheres were separated: one half was post-fixed in 4% PFA for two days and then soaked in cryoprotective solution (30% sucrose in PBS) for immunofluorescence analysis the other further sectioned in different areas (cortex, striatum. cerebellum) quick frozen on dry-ice for western blot, RNA and DNA extraction. Liver specimens were collected similarly. Spleens were collected in PBS for subsequent splenocyte extraction.

## Total RNA-DNA isolation and qRT-PCR for Mecp2 RNA stability, biodistribution and gene expression

Total RNA was isolated from primary neurons and animal tissues (cortex and liver) using the Qiagen RNeasy mini kit (QIAGEN). About 1 µg of RNA was reverse transcribed with random hexamers as primers using ImProm-II Reverse Transcription System (Promega). For quantitative real time PCR (qRT-PCR), Titan HotTaq EvaGreen qPCR mix (BioAtlas) was used and expression levels were normalized respect to β-actin expression. The results were reported as the fold change ($2^{-\Delta\Delta Ct}$) of viral *Mecp2* relative to endogenous *Mecp2*.

The stability of endogenous and viral *Mecp2* RNA was assessed by qRT-PCR (*Supplementary file 3*). The RNA was isolated at the indicated time-points from neurons (WT uninfected and infected with i*Mecp2* vector and *Mecp2$^{-/y}$* infected with i*Mecp2* vector) treated with 10 mg/mL of Actinomycin D (Sigma-Aldrich).

Total DNA was isolated from primary neurons and animal tissues (cortex and liver) using the Qiagen DNeasy Blood and Tissue Kits (QIAGEN). The quantification of vector transgene expression was

calculated by qRT-PCR relative to the endogenous *Mecp2*. The DNA levels were normalized against an amplicon from a single-copy mouse gene, *Lmnb2*, amplified from genomic DNA.

## RiboLace

The RiboLace kit (IMMAGINA Biotechnology S.r.l) was used to isolate from primary neurons (*wild-type* uninfected and infected with i*Mecp2* vector and *Mecp2*$^{-/y}$ infected with i*Mecp2* vector) at DIV 14 two distinct fraction: total RNA and RNA associated to the active ribosomes, according to manufacturer's instructions. Following the isolation, about 100 ng of RNA was reverse transcribed and amplified by qRT-PCR as described above using the oligonucleotide primers to amplify endogenous *Mecp2*, viral *Mecp2* and 18S as housekeeping gene (*Supplementary file 3*). The result was reported as fold change ($2^{-\Delta\Delta Ct}$) in gene expression of viral *Mecp2* relative to endogenous *Mecp2*, in the captured fraction normalized on total RNA.

## CHX-chase analysis

Primary neurons (WT uninfected and *Mecp2*$^{-/y}$ infected with the i*Mecp2* vector) were incubated with CHX (50 µM). The cells were collected after 0 and 8 hr after treatment for protein extraction. Lysate samples were finally prepared for western blot analysis as described below.

## Generation of a MECP2-KO human iPS cell (iPSC) line

Control human iPSC cell line was generated from neonatal primary fibroblasts obtained from ATCC. iPSCs were maintained in feeder-free conditions in mTeSR1 (Stem Cell Technologies) and seeded in HESC qualified matrigel (Corning)-coated 6-well plates. To generate the MECP2-KO cell line, an sgRNA (sgMECP2: 5'-aagcttaagcaaaggaaatc-3') was designed on the third exon of MECP2 using the software crispor.tefor.net. The oligo (Sigma-Aldrich) pairs encoding the 20-nt guide sequences were cloned into the LV-U6-filler-gRNA-EF1α-Blast (*Rubio et al., 2016*). Wild-type human iPSCs were then co-transfected with the LV-U6-sgMECP2-EF1α-Blast and the pCAG-Cas9-Puro using the Lipofectamine Stem Cells Transfection Reagent (ThermoFisher Scientific) (*Giannelli et al., 2018*). Co-transfected colonies were then selected by the combination of puromycin (1 µg/ml, Sigma) and blastidicin (10 µg/ml, ThermoFisher Scientific) and then isolated through single colony picking. Finally, MECP2-KO cell lines were confirmed by Sanger Sequencing and protein absence was further corroborated by immunofluorescence.

Differentiation of human iPSCs in cortical neurons iPSCs were initially differentiated in Neural Progenitors Cells (NPCs) as described (*Iannielli et al., 2018*). NPCs were, then, dissociated with Accutase (Sigma-Aldrich) and plated on matrigel-coated 6-well plates ($3 \times 10^5$ cells per well) in NPC medium. Two days after, the medium was changed with the differentiation medium containing Neurobasal (ThermoFisher Scientific), 1% Pen/Strep (Sigma-Aldrich), 1% Glutamine (Sigma-Aldrich), 1:50 B27 minus vitamin A (ThermoFisher Scientific), 5 µM XAV939 (Sigma-Aldrich), 10 µM SU5402 (Sigma-Aldrich), 8 µM PD0325901 (Tocris Bioscience), and 10 µM DAPT (Sigma-Aldrich) was added and kept for 3 days. After 3 days, the cells were dissociated with Accutase (Sigma-Aldrich) and plated on poly-L-lysine (Sigma-Aldrich)/laminin (Sigma-Aldrich)-coated 12-well plates ($2 \times 10^5$ cells per well) and 24-well plates ($1 \times 10^5$ cells per well) in maturation medium containing Neurobasal (ThermoFisher Scientific), 1% Pen/Strep (Sigma-Aldrich), 1% Glutamine (Sigma-Aldrich), 1:50 B27 minus vitamin A (ThermoFisher Scientific), 25 ng/ml human BDNF (PeproTech), 20 µM Ascorbic Acid (Sigma-Aldrich), 250 µM Dibutyryl cAMP (Sigma-Aldrich), 10 µM DAPT (Sigma-Aldrich) and Laminin for terminal differentiation. At this stage half of the medium was changed every 2–3 days. Viral particles were directly added to cultured neurons after three weeks of differentiation, with a final concentration $10^{10}$ vg/ml. All the analysis was conducted one week after the infection.

## Immunofluorescence

Primary neurons and human iPSCs-derived neurons were fixed with ice-cold 4% paraformaldehyde (PFA) for 30 min at 4°C, washed with PBS (3×) and incubated with 10% donkey serum and 3% Triton X-100 for 1 hr at RT to saturate the unspecific binding site before the overnight incubation at 4°C with the primary antibody. Upon wash with PBS (3×), cells were incubated for 1 hr at RT in blocking solution with DAPI and with Alexa Fluor-488 and Alexa Fluor-594 anti-rabbit or anti-mouse

secondary antibodies. After PBS washes (3×), cells were mounted with fluorescent mounting medium (Dako). Images were captured with a Nikon Eclipse 600 fluorescent microscope.

Tissues were sectioned using cryostat after optimal cutting temperature compound (OCT) embedding in dry ice. Free-floating 50µm-thick coronal sections were rinsed in PBS and were incubated with 10% donkey serum (Sigma-Aldrich) and 3% Triton X-100 (Sigma-Aldrich) for 1 hr at RT to saturate the unspecific binding site before the overnight incubation at 4°C with the primary antibody (diluted in the blocking solution).

Upon wash with PBS (3×), sections were incubated for 1 hr at RT in blocking solution with DAPI (1:1000, Sigma-Aldrich) and with Alexa Fluor-488 and Alexa Fluor-594 anti-rabbit or anti-mouse secondary antibodies (1:1000, ThermoFisher Scientific). After PBS washes (3×), sections were mounted with fluorescent mounting medium (Dako). Confocal images were captured at ×40 or×63 magnification with Leica TCS SP5 Laser Scanning Confocal microscope (Leica Microsystems Ltd).

Cell and tissue where stained with the following primary antibody: rabbit anti-MeCP2 (1:500; Cell Signaling Technology, RRID:AB_2143849), mouse anti-V5 (1:500; ThermoFisher Scientific, RRID:AB_2556564), mouse anti-NeuN (1:300; Merck Millipore, RRID:AB_2298772), rabbit anti-Sox9 (1:500; Sigma-Aldrich, RRID:AB_2239761), rabbit anti-GABA (1:500; Sigma-Aldrich, RRID:AB_477652) anti-chicken_MAP2 (1:500, Abcam, RRID:AB_2138147), anti-OCT4 (1:50, Abcam, RRID:AB_444714).

## Western blot

Protein extracts were prepared in RIPA buffer (10 mM Tris-HCl pH7.4, 150 mM NaCl, 1 mM EGTA, 0.5% Triton and complete 1% protease and phosphatase inhibitor mixture, Roche Diagnostics). Primary neurons, brain and liver lysate samples (50 µg protein lysates) were separated using 8% polyacrylamide gel and then transferred to PVDF membranes. Membranes were incubated overnight at 4°C with the following primary antibodies in 1X PBST with 5% w/v nonfat dry: rabbit anti-MeCP2 (1:1000; Sigma-Aldrich, RRID:AB_262075), mouse anti-V5 (1:1000; ThermoFisher Scientific, RRID:AB_2556564), rabbit anti-pS6 235/236 (1:500, Cell Signaling, RRID:AB_331679), rabbit anti-S6 (1:500, Cell Signaling, RRID:AB_945319), rabbit anti-Calnexin (1:50000, Sigma-Aldrich, RRID:AB_476845), mouse anti-β-Actin (1:50000; Sigma-Aldrich, RRID:AB_262137) or the mice serum (1:200) extracted through retro-orbital bleeding followed by centrifugation (10 min, RT, 13000 rpm). Subsequently, membranes were incubated with the corresponding horseradish peroxidase (HRP)-conjugated secondary antibodies (1:10000; Dako). The signal was then revealed with a chemiluminescence solution (ECL reagent, RPN2232; GE Healthcare) and detected with the ChemiDoc imaging system (Bio-Rad).

## Antibody detection in serum

Serum was extract from mice through retro-orbital bleeding followed by centrifugation (10 min, RT, 13000 rpm). In order to test the sera by immunofluorescence we generated a P19 (RRID:CVCL_2153) *Mecp2*⁻/ʸ cell line using pCAG-spCas9 (*Vierbuchen et al., 2010*) and sg*Mecp2* (*Swiech et al., 2015*). Moreover, isolation of a clone carrying a frameshift mutation (−14nt) in the exon 3 of *Mecp2* gene that ensured ablation of MeCP2 protein (as tested by immunofluorescence). *Mecp2*⁻/ʸ P19 cells were transfected with GFP only (negative control) or with GFP and i*Mecp2*, cells were fixed and incubated in blocking solution as describe above before the overnight incubation at 4°C with the primary antibody mix composed of chicken anti-GFP and either a rabbit anti-MeCP2 (positive control) or the mice serum (1:50) (for details see the above Immunofluorescence paragraph). For western blot assay, the proteins *Mecp2*⁻/ʸ and WT cortices were extracted and separated using 8% polyacrylamide gels. Membranes were incubated overnight at 4°C with a mouse anti-MeCP2 (1:1000; Sigma-Aldrich, RRID:AB_262075, positive control) or the mice serum (1:200) (for details see the western blot paragraph).

## Fluorescent intensity measurements

Brain sections were processed for immunolabeling as above and confocal images were captured at ×63 magnification with Leica TCS SP5 Laser Scanning Confocal microscope (Leica Microsystems Ltd) using the identical settings. Then, the quantification of the signal was performed using ImageJ software (NIH, US). The fluorescent signal was measured as described (*Iannielli et al., 2018*).

## Spleen cell isolation

Spleens were triturated in PBS and cell pelleted (7 min, 1500 rpm) to be incubated in ACK buffer (5 min, RT) to lyse blood cells. The reaction was stopped diluting 1:10 the ACK buffer in PBS and removing it by centrifugation (7 min, 1500 rpm). Cells were than counted and frozen in FBS:DMSO (9:1 ratio) solution.

## Flow cytometry

Spleen cells were incubated with 25 µl of Ab mix listed in *Supplementary file 3* for 30' at 4°. Red blood cells lysis was performed with BD Phosflow (BD Bioscience, 558049) according to manufacturer's instruction. Labeled cells were washed two times with PBS 1% FBS and analysed with a BD LSRFortessa analyser, results were analysed with FlowJo 10 software.

## T cell proliferation

Spleen cells were labeled with Cell Proliferation Dye eFluor 670 (eBioscience, CA, USA) according to manufacturer's instructions and stimulated with $10^4$ bone-marrow derived DC transduce with lentiviral vector encoding for *Mecp2* (10:1, T:DC) in RPMI 1640 medium (Lonza, Switzerland), with 10% FBS (Euroclone, ECS0180L), 100 U/ml penicillin/streptomycin (Lonza, 17-602E), 2 mM L-glutamine (Lonza, 17-605E), Minimum Essential Medium Non-Essential Amino Acids (MEM NEAA) (GIBCO, 11140–035), 1 mM Sodium Pyruvate (GIBCO, 11360–039), 50 nM 2-Mercaptoethanol (GIBCO, 31350–010). Alternatively, spleen cells were stimulated with anti-CD3e monoclonal Ab (BD Bioscience, 553058) (1 µg/mL). After 4 days, T cells were collected, washed, and their phenotype and proliferation were analysed by flow cytometry. Antibodies used for flow cytometry are listed in *Supplementary file 4*.

## Elispot assays

$CD8^+$ T cells were magnetically isolated from the spleen (Miltenyi Biotec, 130-104-075). $10^5$ $CD8^+$ T cells were plated in triplicate in ELISPOT plates (Millipore, Bedford, MA) pre-coated with anti–IFN-γ capture monoclonal Ab (2.5 µg/mL; BD Pharmingen, R46A2) in the presence of IL-2 (50 U/mL; BD Pharmingen) and $10^5$ irradiated (6000 rad) un-transduced or LV.*Mecp2*-transduced autologous EL-4 cells. After 42 hr of incubation at 37°C 5% $CO_2$, plates were washed and IFN-γ–producing cells were detected by biotin-conjugated anti–IFN-γ monoclonal Ab (0.5 µg/mL; BD Pharmingen, XMG 1.2). Streptavidin-HRP conjugate (Roche) was added. Total splenocytes or total BM (0,35 × $10^6$ cells/well) were plated in complete RPMI in triplicate in ELISPOT plates pre-coated with rhIDUA (2 µg/well). After 24 hr of incubation at 37°C 5% $CO_2$, plates were washed and anti-IDUA IgG secreting cells were detected with peroxidase-conjugated rabbit anti–mouse immunoglobulin (Sigma-Aldrich, RRID:AB_258008). All plates were reacted with $H_2O_2$ and 3-Amino-9-ethylcarbazole (Sigma-Aldrich, RRID:AB_2767485). Spots were counted by ImmunoSpot reader (Cellular Technology Limited).

## Computational analysis

FASTQ reads were quality checked and trimmed with FastQC (*Wingett and Andrews, 2018*). High quality trimmed reads were mapped to the mm10/GRCm38.p6 reference genome with Bowtie2 v2.3.4.3 (*Langmead and Salzberg, 2012*). Gene counts and differential gene expression were calculated with featureCount using latest GENCODE main annotation file (*Liao et al., 2013*) and DESeq21 respectively *Love et al. (2014)*. Geneset functional enrichment was performed with GSEA (*Subramanian et al., 2005*). Downstream statistics and Plot drawing were performed with R. Heatmaps were generated with GENE-E (The Broad Institute of MIT and Harvard).

## Rotarod

Mice were assessed on an accelerating rotarod (Ugo-Basile, Stoelting Co.). Revolutions per minute (rpm) were set at an initial value of 4 with a progressive increase to a maximum of 40 rpm across the 5 min test session. The animals were given two days of training and one day of test, each session consisting of three trials. Latency to fall was measured by the rotarod timer.

## Elevated plus maze (EPM)

The test uses an elevated cross (+) apparatus with two open and two enclosed arms (length 45 cm): the open arms had low walls (0,5 cm) while the closed arms had high walls (20 cm). The mouse was placed in the center of the apparatus and it could move and explore freely the maze for 10 min. The amount of the time that the animal spent in closed arms were measured and analyzed with EthoVision XT system. The maze was cleaned with water and 70% ethanol before the next mouse was placed on the apparatus.

## Statistics

Values are expressed as mean ± standard deviation as indicated. All statistical analysis was carried out in GraphPad Prism 8.0, using one-way ANOVA, two-way ANOVA, Mantel-Cox test (survival curves) and non-parametric Mann-Whitney U test (two-tailed) where unpaired t-test was applied. P-values below 0.05 were considered significant. In multi-group comparisons, multiple testing corrections for pairwise tests among groups was applied using Tukey's post hoc analysis.

## Acknowledgements

We thank N Landsberger, E Cattaneo, F Ciceri and L Naldini for providing valuable reagents and mouse strains. We are grateful to D Bonanomi, S Biffo and all members of the Broccoli's lab for helpful discussion. We acknowledge the FRACTAL and ALEMBIC core facilities for expert supervision in flow-cytometry and confocal imaging, respectively. This work was supported by the Telethon Foundation grant (GGP19038) to VB.

## Additional information

### Funding

| Funder | Grant reference number | Author |
|---|---|---|
| Fondazione Telethon | GGP19038 | Vania Broccoli |

The funders had no role in study design, data collection and interpretation, or the decision to submit the work for publication.

### Author contributions

Mirko Luoni, Conceptualization, Data curation, Formal analysis, Validation, Visualization, Methodology; Serena Giannelli, Conceptualization, Data curation, Formal analysis, Supervision, Validation, Investigation, Visualization, Methodology; Marzia Tina Indrigo, Formal analysis, Investigation, Visualization, Methodology; Antonio Niro, Laura Passeri, Data curation, Formal analysis, Visualization, Methodology; Luca Massimino, Formal analysis, Visualization, Methodology; Angelo Iannielli, Formal analysis, Investigation; Fabio Russo, Data curation, Investigation, Visualization, Methodology; Giuseppe Morabito, Data curation, Investigation, Methodology; Piera Calamita, Benjamin Deverman, Methodology; Silvia Gregori, Conceptualization, Data curation, Formal analysis, Validation, Investigation, Visualization, Methodology; Vania Broccoli, Conceptualization, Resources, Data curation, Formal analysis, Supervision, Funding acquisition, Validation, Investigation, Visualization, Methodology, Project administration

### Author ORCIDs

Mirko Luoni https://orcid.org/0000-0002-5006-1827
Piera Calamita http://orcid.org/0000-0002-9029-9346
Benjamin Deverman http://orcid.org/0000-0002-6223-9303
Vania Broccoli https://orcid.org/0000-0003-4050-0926

## Ethics

Animal experimentation: All procedures were performed according to protocols approved by the internal IACUC and reported to the Italian Ministry of Health according to the European Communities Council Directive 2010/63/EU.

## Decision letter and Author response

Decision letter https://doi.org/10.7554/eLife.52629.sa1
Author response https://doi.org/10.7554/eLife.52629.sa2

# Additional files

## Supplementary files

• Supplementary file 1. Quantifiation of Mecp2 gene transfer in infected brain tissue. Quantitative assessment of transduced (V5$^+$) astrocytes (Sox9+) and neurons (NeuN+) in the cerebral cortical tissue at each different i*Mecp2* viral dose.

• Supplementary file 2. No evidence of liver toxicity in mice administered with high AAV dose. Blood serum levels of liver enzyme and liver histochemical analysis (representative images) were used as indicators of liver health. # Reference values for C57BL/6J male mice were taken from the mouse phenome database https://phenome.jax.org/. Abbreviations, Treat: treatment, ALB: albumin, ALP: Alkaline phosphatase, ALT: Alanine aminotransferase, HE: hematoxylin/eosin staining. All values are indicated as mean ± SD, n = 3, SD. Scale bar: 200 µm.

• Supplementary file 3. Primers employed for qRT-PCRs.

• Supplementary file 4. List of antibodies for flow cytometry.

• Transparent reporting form

## Data availability

Sequencing data have been deposited in GEO under accession code GSE125155.

The following dataset was generated:

| Author(s) | Year | Dataset title | Dataset URL | Database and Identifier |
|---|---|---|---|---|
| Luoni M, Giannelli S, Massimino L, Broccoli V | 2020 | RNA-Seq MecP2 | https://www.ncbi.nlm.nih.gov/geo/query/acc.cgi?acc=GSE125155 | NCBI Gene Expression Omnibus, GSE125155 |

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
