## [Decision Letter]

**Acceptance summary:**

We fully appreciate the value of this work showing how the use of novel viral vectors can restore physiological levels of Mecp2, a gene which mutation is causal to Rett Syndrome. We also wish to stress that the addition to the revised version have addressed most of the points raised by the reviewers and significantly improved the strength of the claims put forward by the authors.

**Decision letter after peer review:**

[Editors’ note: the authors submitted for reconsideration following the decision after peer review. What follows is the decision letter after the first round of review.]

Thank you for submitting your work entitled "Whole brain delivery of an instability-prone Mecp2 transgene ameliorates deficits in mouse models of Rett syndrome" for consideration by *eLife*. Your article has been reviewed by two peer reviewers, and the evaluation has been overseen by a Reviewing Editor and a Senior Editor. The reviewers have opted to remain anonymous.

Our decision has been reached after consultation between the reviewers. Based on these discussions and the individual reviews below, we regret to inform you that your work will not be considered further for publication in *eLife*.

Both reviewers have stressed the quality and interest of your study aimed at restoring in mice the physiological levels of MecP2, a gene which mutation is causal to Rett syndrome. While the reviewers acknowledged the quality of your work and the thoroughness of the experimental approaches, they raised two main concerns that prevents us from being fully supportive for publication of your manuscript.

1) Main point: several publications (Hordeaux et al., 2018; Matsuzaki et al., 2018; Liguore et al., 2019) highlight the limited capacity of the AAV-PHP.B/eB viruses to efficiently infect brain cells in primates, thus impacting the translational impact of the study. At least one pf the reviewers felt that this made your work potentially irrelevant for translation into humans. If you feel that he/she is incorrect, this issue would need to be addressed.

2) The study should include an expanded analysis of female heterozygotes as pointed out by the two reviewers.

If you were able respond to these two main issues, we would be happy to reconsider a revised version of your manuscript.

Reviewer #1:

The manuscript by Luoni et al., highlights observations stemming from pre-clinical gene therapy efforts conducted in a mouse model of the neurodevelopmental condition Rett syndrome. The author's show phenotypic improvement, rescue of gene expression alterations, improvements in hypoactive protein synthesis pathways in treated mice, and identify and address an immune response complication seen in male mice treated with higher doses of rescue virus. While the results on the whole are very interesting, add to existing work in the area, and further establish the phenotypes of Rett syndrome mice are correctable by functional Mecp2 gene reintroduction, there is a significant issue that lessens my enthusiasm for this work to be published in *eLife*.

Major Issue:

The premise of the study is a pre-clinical assessment of peripherally-administered gene therapy using an improved AAV serotype that could bypass some of the major hurdles currently hindering clinical trial development in the field. Unfortunately, and likely while the study was in progress, the PHP.B series of viral serotypes were found to cross the BBB efficiently only in species (or strains) expressing a specific Ly6A haplotype that is not present in primates. This indicates PHP.eB likely has no clinical utility in humans unless directly infused into CNS, and thus offers no clear advantage over other current gene therapy serotypes such as AAV9. This significantly decreases the impact for the lack of peripheral organ toxicity, high efficiency of CNS transduction when peripherally administered, and potentially the efficiency of brain expression that are reported here. The work itself is well conceived and executed (notwithstanding some of the less critical comments below), and would still be of interest to the field. But the limits of the PHP.eB system to experimental contexts lessens the impact of the results.

*Reviewer #2:*

In this manuscript by Luoni et al., the authors perform a proof of concept for the use of AAV- PHP.eb viruses, which were shown to cross the blood-brain barrier in mice, to restore physiologically relevant levels of Mecp2, a gene which mutation is causal to Rett syndrome. Because high levels of Mecp2 can also induce deficits in brain development and function, the authors set out to explore different constructs including one that drive the expression of a relatively unstable mRNA (iMecp2). Using an elegant combination of cell cultures and in vivo injections, the authors show: (i) that they can drive a broad expression of Mecp2 in the brain; (ii) that the expression levels, assessed by western blot and immunostainings, can restore Mecp2 expression levels in mutant conditions. They further tested several viral doses of iMecp2 to test the rescue in male mutants (Mecp2y/-), which harbor a severe phenotype, but do not reflect the pathological condition (Females Mecp2-/+). They found that increasing doses were progressively rescuing behavior deficits and expression levels, except at high dose, where they found that the viral injections triggered an immune response, specifically in mutants, probably linked to a role of Mecp2 in regulatory T cells. By using cyclosporine treatment to the high dose viral injection, the authors could trigger a robust rescue at all levels. The authors also show a mild albeit significant improved rescue in female Mecp2+/- (Figure 7). Finally using RNA-seq, the authors show in Mecp2y/- males that iMecp2 injections rescued most of the transcriptional defects normally associated with Mecp2 deficiency as well as established that this viral infection does not major changes in control males and females.

Collectively, this study includes an impressive and highly convincing set of experiments that are well presented and support the main claims of the authors. The expression rate and levels obtained by the viral transfection are quite remarkable, the controls extremely solid and the effects observed will be of interest for both neurobiologists and clinicians. However, in its present form, several issues need to be addressed before the article is suitable for publication.

Major points:

- Because this article is really designed and presented as a proof of concept for therapy, the females should be presented in a main figure (instead of Figure 5) and should be stressed in the analysis. Is there any difference in transduction/expression levels in males vs females? Is there an immune response as well?

- While the cyclosporine A treatment has been performed on iMecp2 transfected mutants, there does not seem to have a control group. The authors should provide either a clear reason to discard this condition since it would change the statistical power of their analyses.

---

## [Author Response]

[Editors’ note: The authors appealed the original decision. What follows is the authors’ response to the first round of review.]

Both reviewers have stressed the quality and interest of your study aimed at restoring in mice the physiological levels of MecP2, a gene which mutation is causal to Rett syndrome. While the reviewers aknowledged the quality of your work and the thoroughness of the experimental approaches, they raised two main concerns that prevents us from being fully supportive for publication of your manuscript.1) Main point: several publications (Hordeaux et al., 2018; Matsuzaki et al., 2018; Liguore et al., 2019) highlight the limited capacity of the AAV-PHP.B/eB viruses to efficiently infect brain cells in primates, thus impacting the translational impact of the study. At least one pf the reviewers felt that this made your work potentially irrelevant for translation into humans. If you feel that he/she is incorrect, this issue would need to be addressed.

Since RTT is a severe neurological disorder involving many brain areas, the use of viral vectors able to efficiently transduce the brain is required to investigate the benefit of a potential gene therapy strategy. Currently, the AAV-PHP.eB represents the main technology to achieve this aim, due to its efficient whole brain delivery, at least in mouse models, where a critical evaluation of the gene therapy effects is still missing. Indeed, recent works based on the AAV9 have shown that the gene therapy efficacy during the symptomatic stage is low (Gadalla et al., 2017; Sinnett et al., 2017). Indeed the same authors claimed that: “…While AAV9 appears to be insufficiently efficient in terms of brain transduction after systemic delivery of MECP2 to achieve the desired therapeutic benefit, combining the safer second-generation cassette together with capsids with improved brain penetrance may effectively pair effective CNS gene transfer with safe levels of peripheral MeCP2 transgene expression]…” (Gadalla et al., 2017). Thus, the use of the AAV-PHP.eB allowed us to overcome this important limitation, validating the strong potential of gene therapy also in symptomatic mice, that closely resembles the pathological condition of RTT patients. The reviewers are right in emphasizing that the engineered synthetic AAV-PHP.eB capsid has not the same efficiency in nonhuman primates as in mice to transduce the brain upon systemic intra-vascular delivery (Hordeaux et al., 2018). Exactly this same subject has been highlighted also in our discussion since it’s a hot topic in the field. However, considering the three arguments explicated below our study should be considered a fundamental step to develop an efficient and safe therapy for RTT which necessitates of a brainwide gene therapy approach.

1) Following the seminal discovery of the PHP.eB capsid family, large projects in academics and companies have been carried out to identify and characterize similar engineered PHP.eB-like capsids with conserved brain transduction efficiency from mouse to non-human primates. Indeed, participating to the annual meeting of the American Society of Gene and Cell Therapy (ASGCT) in early May this year, I have personally attended some presentations that described the generation of new PHP.eB-like capsids with the ability to cross the blood-brain barrier and transduce the brain in non-human primates.

Thus, it can be anticipated that these novel viral capsids will become available to the community in a short while. As for instance:

Oral Presentation – Abstract #102. Engineered AAVS for CNS Transduction and Peripheral Organ DeTargeting across Species after Systemic Delivery. Nicholas Flytzanis^1^, Nick Goeden^1^, James Pickel^2^, Viviana Gradinaru^1. 1^BBE, California Institute of Technology, Pasadena, CA, ^2^NIMH, National Institutes of Health, Bethesda, MD.

Through this method, we’ve identified a panel of novel engineered capsids capable of high levels of production, extremely efficient transduction of the central nervous system (CNS) after intravenous delivery, and negligible presence throughout the rest of the body. Notably, AAV9.DT-N exhibits the highest transduction of neurons in the brain of any engineered variant to date, with less than 1% of cells in the liver transduced, a decrease of over 100-fold from AAV9. Most importantly, these capsids’ ability to efficiently cross the blood-brain barrier and transduce CNS neurons is not limited to the rodents they were selected in, having also been validated in non-human primates.

Oral Presentation Abstract #48. Targeted In Vivo Biopanning of AAV Capsid Libraries Using Cell TypeSpecific RNA Expression. Nonnenmacher et al.

.…. Interestingly, several variants highly similar to the PHP.eB capsid were recovered, suggesting that our method allowed a rapid selection of high-performance capsids. Top-ranking capsids were then individually tested, and several variants showed CNS transduction similar to or higher than the PHP.eB benchmark. Through this method, we’ve identified a panel of novel engineered capsids capable of high levels of production, extremely efficient transduction of the central nervous system (CNS) after intravenous delivery, and negligible presence throughout the rest of the body both in mice and non-human primates.

2) Moreover, our viral vector design and administration route have a strong relevance to define safety and efficacy of a systemic gene therapy approach for RTT. These important issues need to be addressed and delved as much as possible in pre-clinical settings. PHP.eB capsid has given us the possibility to investigate this matter in a way that no other natural capsid as ever displayed.

3) Finally, we strongly believe that translating a therapy from a murine model to the human system is long a process that cannot be limited to a single choice. All the possible settings have to be carefully evaluated and weighted in order to develop the best therapeutic treatment. In this perspective, our contribute can be seminal in this process.

2) The study should include an expanded analysis of female heterozygotes as pointed out by the two reviewers.

Most of our work is executed in male, and not female RTT mice, since only these animals recapitulate the cardinal features of the human RTT phenotype including the severe motor and cognitive deficits (Guy et al., 2001; Chen et al., 2001). Whereas heterozygous female mice have mild deficits that appear only starting from 8-10 months of age. Moreover, these symptoms are highly variable and more difficult to evaluate than those of null male mice. Indeed, most of the studies aimed to evaluate the effect of Mecp2 restoration mediated by genetic or viral strategies have been conducted in male RTT mice (Guy et al., 2007; Robinson et al., 2012; Gadalla et al., 2013; Gadalla et al., 2017; Sinnett et al., 2017). Nevertheless, we can further expand our analyses in heterozygous female mice comparing the transduction profile between males and females animals after the treatment as requested by the two reviewers.

Reviewer #1:The manuscript by Luoni et al., highlights observations stemming from pre-clinical gene therapy efforts conducted in a mouse model of the neurodevelopmental condition Rett syndrome. The author's show phenotypic improvement, rescue of gene expression alterations, improvements in hypoactive protein synthesis pathways in treated mice, and identify and address an immune response complication seen in male mice treated with higher doses of rescue virus. While the results on the whole are very interesting, add to existing work in the area, and further establish the phenotypes of Rett syndrome mice are correctable by functional Mecp2 gene reintroduction, there is a significant issue that lessens my enthusiasm for this work to be published in eLife.Major issue:The premise of the study is a pre-clinical assessment of peripherally-administered gene therapy using an improved AAV serotype that could bypass some of the major hurdles currently hindering clinical trial development in the field. Unfortunately, and likely while the study was in progress, the PHP.B series of viral serotypes were found to cross the BBB efficiently only in species (or strains) expressing a specific Ly6A haplotype that is not present in primates. This indicates PHP.eB likely has no clinical utility in humans unless directly infused into CNS, and thus offers no clear advantage over other current gene therapy serotypes such as AAV9. This significantly decreases the impact for the lack of peripheral organ toxicity, high efficiency of CNS transduction when peripherally administered, and potentially the efficiency of brain expression that are reported here. The work itself is well conceived and executed (notwithstanding some of the less critical comments below), and would still be of interest to the field. But the limits of the PHP.eB system to experimental contexts lessens the impact of the results.Reviewer #2:In this manuscript by Luoni et al., the authors perform a proof of concept for the use of AAV- PHP.eb viruses, which were shown to cross the blood-brain barrier in mice, to restore physiologically relevant levels of Mecp2, a gene which mutation is causal to Rett syndrome. Because high levels of Mecp2 can also induce deficits in brain development and function, the authors set out to explore different constructs including one that drive the expression of a relatively unstable mRNA (iMecp2). Using an elegant combination of cell cultures and in vivo injections, the authors show: (i) that they can drive a broad expression of Mecp2 in the brain; (ii) that the expression levels, assessed by western blot and immunostainings, can restore Mecp2 expression levels in mutant conditions. They further tested several viral doses of iMecp2 to test the rescue in male mutants (Mecp2y/-), which harbor a severe phenotype, but do not reflect the pathological condition (Females Mecp2-/+). They found that increasing doses were progressively rescuing behavior deficits and expression levels, except at high dose, where they found that the viral injections triggered an immune response, specifically in mutants, probably linked to a role of Mecp2 in regulatory T cells. By using cyclosporine treatment to the high dose viral injection, the authors could trigger a robust rescue at all levels. The authors also show a mild albeit significant improved rescue in female Mecp2+/- (Figure 7). Finally using RNA-seq, the authors show in Mecp2y/- males that iMecp2 injections rescued most of the transcriptional defects normally associated with Mecp2 deficiency as well as established that this viral infection does not major changes in control males and females.Collectively, this study includes an impressive and highly convincing set of experiments that are well presented and support the main claims of the authors. The expression rate and levels obtained by the viral transfection are quite remarkable, the controls extremely solid and the effects observed will be of interest for both neurobiologists and clinicians. However, in its present form, several issues need to be addressed before the article is suitable for publication.Major points:- Because this article is really designed and presented as a proof of concept for therapy, the females should be presented in a main figure (instead of Figure 5) and should be stressed in the analysis. Is there any difference in transduction/expression levels in males vs females? Is there an immune response as well?

We will be happy to compare the viral transduction efficiency and expression levels of the transgene in Mecp2 mutant males vs females.

In Mecp2 mutant females as well as wild-type mice we did not detect any notable immune response to the viral transgene. These data are in line with our hypothesis that only Mecp2 null mice have an exacerbated immune response due to an impaired immune regulatory system as initially described by Li et al., (2014).

- While the cyclosporine A treatment has been performed on iMecp2 transfected mutants, there does not seem to have a control group. The authors should provide either a clear reason to discard this condition since it would change the statistical power of their analyses.

We did not present the results of the Mecp2 mutant animals injected only with cyclosporine A since they are fully overlapping with those in mice treated both with the GFP expressing AAV and cyclosporine A. However, we can present these data in the revised version.

[Editors’ note: The authors submitted a revised manuscript for consideration. What follows is the authors’ response to remaining comments from the first round of review.]

Major issue:The premise of the study is a pre-clinical assessment of peripherally-administered gene therapy using an improved AAV serotype that could bypass some of the major hurdles currently hindering clinical trial development in the field. Unfortunately, and likely while the study was in progress, the PHP.B series of viral serotypes were found to cross the BBB efficiently only in species (or strains) expressing a specific Ly6A haplotype that is not present in primates. This indicates PHP.eB likely has no clinical utility in humans unless directly infused into CNS, and thus offers no clear advantage over other current gene therapy serotypes such as AAV9. This significantly decreases the impact for the lack of peripheral organ toxicity, high efficiency of CNS transduction when peripherally administered, and potentially the efficiency of brain expression that are reported here. The work itself is well conceived and executed (notwithstanding some of the less critical comments below), and would still be of interest to the field. But the limits of the PHP.eB system to experimental contexts lessens the impact of the results.

The reviewer is right in emphasizing that the engineered synthetic AAV-PHP.eB capsid has not the same efficiency in non-human primates as in mice to transduce the brain upon systemic intra-vascular delivery (Hordeaux et al., 2018). This very subject has been highly regarded also in our discussion since it’s a hot topic in the field. However, given the considerations that we argument below, our study should be considered a fundamental step in order to develop an efficient and safe therapy for RTT. This an important goal, since a brain-wide gene therapy is the only clinical option for an efficient treatment of this disease.

Following the seminal discovery of the PHP.B capsid family, large projects in academics

(Church, Gradinaru, Maguire, Gray, Buening labs) and companies (Voyager, Spark Ther., Passage Bio) have been carried out to identify and characterize similar engineered AAV9like capsids with conserved brain transduction efficiency from mouse to non-human primates and preliminary data have been presented to international meetings in the last 12 months (ASGCT-Washington 5/2019, ESGCT-Barcelona 10/2019). Thus, it can be anticipated that these novel viral capsids will be published and become available to the community in a short while. On this line, the Maguire’s lab has already published a new engineered AAV9 capsid, named AAV-F, with a new 7-aminoacid peptide in the exact same site were PHP.B capsid has been engineered, corresponding to position 588-589 of VP1 aminoacidic sequence (Hanlon et al., 2019). However, differently from the PHP.B, the AAV-F capsid is able to efficiently transduce the brain tissue after systemic delivery in BALB/c mice where the PHP.B receptor (Ly6A) is mutated and unfunctional. This finding suggests that the AAV-F crosses the brain endothelium through a different membrane receptor, that if conserved in evolution, might provide AAV-F the ability to transduce non-human primates and humans as well. Additionally, the Gradinaru’s lab has reported the characterization of two new PHP.B-like engineered capsids, named AAV-CAP.b10 and b22 that after systemic delivery in adult marmosets can transduce brain neurons 15-fold and 17-fold higher than AAV9, respectively (ASGCT 2020, Abstract). Thus, new AAV9 engineered capsids will be soon available with similar operative properties of the PHP.B, but with a strong translational potential for gene therapy applications in humans.

In this perspective, our viral vector design and administration route has a strong relevance to define the safety and efficacy of a systemic gene therapy approach for RTT. The results obtained in our study with the PHP.eB remain absolutely valid and, therefore, can accelerate the use of the new engineered capsid compatible with translational applications. PHP.eB capsid has given us the possibility to investigate this matter in a way that no other natural capsid would allow.

Reviewer #2:In this manuscript by Luoni et al., the authors perform a proof of concept for the use of AAV- PHP.eb viruses, which were shown to cross the blood-brain barrier in mice, to restore physiologically relevant levels of Mecp2, a gene which mutation is causal to Rett syndrome. Because high levels of Mecp2 can also induce deficits in brain development and function, the authors set out to explore different constructs including one that drive the expression of a relatively unstable mRNA (iMecp2). Using an elegant combination of cell cultures and in vivo injections, the authors show: (i) that they can drive a broad expression of Mecp2 in the brain; (ii) that the expression levels, assessed by western blot and immunostainings, can restore Mecp2 expression levels in mutant conditions. They further tested several viral doses of iMecp2 to test the rescue in male mutants (Mecp2y/-), which harbor a severe phenotype, but do not reflect the pathological condition (Females Mecp2-/+). They found that increasing doses were progressively rescuing behavior deficits and expression levels, except at high dose, where they found that the viral injections triggered an immune response, specifically in mutants, probably linked to a role of Mecp2 in regulatory T cells. By using cyclosporine treatment to the high dose viral injection, the authors could trigger a robust rescue at all levels. The authors also show a mild albeit significant improved rescue in female Mecp2+/- (Figure 7). Finally using RNA-seq, the authors show in Mecp2y/- males that iMecp2 injections rescued most of the transcriptional defects normally associated with Mecp2 deficiency as well as established that this viral infection does not major changes in control males and females.Collectively, this study includes an impressive and highly convincing set of experiments that are well presented and support the main claims of the authors. The expression rate and levels obtained by the viral transfection are quite remarkable, the controls extremely solid and the effects observed will be of interest for both neurobiologists and clinicians. However, in its present form, several issues need to be addressed before the article is suitable for publication.

We are grateful to the reviewer for the positive evaluation and comments on our manuscript.

Major points:- Because this article is really designed and presented as a proof of concept for therapy, the females should be presented in a main figure (instead of Figure 5) and should be stressed in the analysis. Is there any difference in transduction/expression levels in males vs females? Is there an immune response as well?

We agree with the reviewer that Mecp2 mutant heterozygous females represent a crucial part for a proof of concept of gene therapy for RTT, since they are the genetic model of the disease. For this reason, we moved these data to a principal figure (now Figure 7). Nevertheless, both the mild phenotype and the variability of heterozygous females do not recapitulate the strong neurological deficits affecting RTT patients. In fact, only hemizygous mutant male mice present severe neuropathological symptoms resembling somehow the patient condition. For these reasons, we consider that results obtained both in male and female mice are important to be presented in the main figures.

Concerning the first question, a direct comparison of transduction/expression levels between males and females cannot be performed since the experimental conditions tested in this work are intrinsically different. In fact, females were injected after 5 months of life and sacrificed after around 11 months post-treatment, while males received the same amount of viral vectors after 4 weeks of age and they were euthanized about 2 months post-treatment. Thus, either the weight at the time of the injection (27 grams median weight of 5-months old Mecp2 heterozygous female vs 12 grams median weight of 4-weeks old Mecp2 KO animals) and the time of the final analyses were significantly different. Having said that, we completely agree with the reviewer that a more detailed characterization of the treated females is required to strengthen our work. For this reason, we have analyzed the levels of transduction and expression of our transgene also in this group of mice and included these results in the Supplemental Figure 6. As described in the revised manuscript, we observed that Mecp2 levels in the cortical tissue were higher in treated females in comparison with WT untreated mice (males and females) or control females (Supplementary Figure 7A). Conversely, in the liver we detected only a mild increase of both transduction and expression levels in comparison with the control group (Supplementary Figure 7A), indicating a reduced viral persistence. Loss of AAV genomes after 11 months from the injection is probably a consequence of small but continued proliferation of the liver cells, a general phenomenon which affects AAV transduction in this organ (i.e. Dane et al., 2009). Moreover, we have included in the revised manuscript a comprehensive characterization of the viral transduction profile in the brain tissue of this group of mice, showing a similar distribution of transduced neurons and astrocytes between males and females (Supplementary Figure 7B).

Concerning the second question, in Mecp2 mutant females as well as wild-type mice we did not detect any notable immune response to the viral transgene. These data are in line with our hypothesis that only Mecp2 null mice have an exacerbated immune response due to an impaired immune regulatory system as initially described by Li et al., (2014).

- While the cyclosporine A treatment has been performed on iMecp2 transfected mutants, there does not seem to have a control group. The authors should provide either a clear reason to discard this condition since it would change the statistical power of their analyses.

Originally, we did not present the results of the Mecp2 mutant animals injected only with cyclosporine A since they are fully overlapping with those in mice treated both with the GFP expressing AAV and cyclosporine A. In line with the reviewer request, we added the analysis with this specific treatment in the revised Figure 5A.